# AIDBench: A benchmark for evaluating the authorship identification capability of large language models

## Abstract

As large language models (LLMs) rapidly advance and integrate into daily life, the privacy risks they pose are attracting increasing attention. We focus on a specific privacy risk where LLMs may help identify the authorship of anonymous texts, which challenges the effectiveness of anonymity in real-world systems such as anonymous peer review systems. To investigate these risks, we present AID-Bench, a new benchmark that incorporates several author identification datasets, including emails, blogs, reviews, articles, and research papers. AIDBench utilizes two evaluation methods: one-to-one authorship identification, which determines whether two texts are from the same author; and one-to-many authorship identification, which, given a query text and a list of candidate texts, identifies the candidate most likely written by the same author as the query text. We also introduce a Retrieval-Augmented Generation (RAG)-based method to enhance the large-scale authorship identification capabilities of LLMs, particularly when input lengths exceed the models' context windows, thereby establishing a new baseline for authorship identification using LLMs. Our experiments with AIDBench demonstrate that LLMs can correctly guess authorship at rates well above random chance, revealing new privacy risks posed by these powerful models.

## 1 Introduction

Large language models (LLMs) have experienced widespread application (Zhao et al., 2023; Chang et al., 2024) due to their remarkable capabilities in processing and generating human-like text. Specifically, they are able to follow human instructions (Aw et al., 2023) to accomplish various tasks, both in professional and personal spheres. At the same time, the wide adoption of LLMs also brings impact on societal paradigms that cannot be overlooked (Huang et al., 2023; Chen et al., 2024).

This paper delves into one such challenge posed by LLMs: the issue of privacy risks (Yang et al., 2013; Zheng et al., 2024; Li et al., 2023a;c). In the study of privacy risks associated with language models, researchers primarily focus on the leakage of private information in the training data (Li et al., 2023b; Nasr et al., 2023; Wen et al., 2023). As we navigate through a world where information is increasingly digitized and interconnected, the potential for privacy breaches becomes more diverse and realistic. Specifically, we explore how large models, through their advanced capabilities and extensive knowledge base, can identify anonymous text based on the context and content at hand. Due to exceptional text analysis capabilities, LLMs can be used to infer private information and considered as a tool to help individuals conduct privacy attacks (Staab et al., 2023) and infer the author's information of anonymous texts (Nyffenegger et al., 2023).

This kind of privacy breach is well motivated by the real-world application of anonymous systems. Systems such as academic peer reviews[1], student confession websites, and corporate employee anonymous communication platforms, are designed to protect the identity of individuals (Edman & Yener, 2009; Hohenberger et al., 2014), ensuring that their comments or concerns can be shared without fear of retribution or bias. However, the integrity of these systems is highly dependent on

---

[1] https://openreview.net

the effectiveness of their anonymity. The anonymity of these systems is under threat if the text produced within them can be used to accurately identify the author. For instance, consider the academic peer review system, where anonymity is a cornerstone of the process (Panadero & Alqassab, 2019; Coomber & Silver, 2010). Typically, within a conference, a reviewer may be assigned with evaluating multiple papers, and the reviewer is given a random identifier when they write a review for a paper (Ragone et al., 2013; Zhang et al., 2022b). A reviewer may participate in multiple events over a period of time. People usually cannot link the random identifiers of a single reviewer in nature. However, equipped with powerful large language models, one may be able to identify a reviewer based on the careful comparison between the content of a review and that of other reviews. If the above process were achieved with high probability, it would pose a significant de-anonymization risk in current systems.

In literature, a closely related problem is authorship attribution (AA) (Neal et al., 2017; Stamatatos, 2009; Barlas & Stamatatos, 2020b). Usually, there is a list of suspects and a collection of texts with confirmed authorship for each individual on the list. The objective is to determine the author of texts whose authorship is in question, a task that can be divided into three distinct categories: closed-set attribution where the list of suspects is assumed to contain the actual author of the disputed texts, open-set attribution where the true author may not be included in the initial list of suspects, and author verification where the disputed text is compared against a single candidate author and verify whether this one individual is indeed the author of the text in question. Specifically, Huang et al. (2024a) employ LLMs to perform authorship attribution with carefully designed prompts on two datasets open-source blog (Schler et al., 2006) and email datasets (Klimt & Yang, 2004). A recent survey Huang et al. (2024b) summarizes recent progress and available datasets for authorship attribution, and extend the landscape to LLM-generated text detection and attribution. Moreover, it is worthy to mention that PAN is a series of scientific events with datasets and competitions focusing on digital text forensics and stylometry, specifically cross-topic and cross-genre authorship verification (Stamatatos et al., 2022), author identification and multi-author analysis (Bevendorff et al., 2023), and multi-author writing style analysis and generative AI authorship verification (Bevendorff et al., 2024). However, these setups differ significantly from the challenging anonymous review systems where one has to compare hundreds of texts without effective author profiles available.

In this paper, we introduce AIDBench, a comprehensive benchmark designed for extensive and in-depth authorship identification. The goal is to identify texts written by the same author when a specific text is known to belong to that person; specifically, given a disputed text, our task is to find candidate texts most likely authored by the same individual. To systematically study this problem, we collect multiple datasets, including research papers, Enron emails (Klimt & Yang, 2004), blogs (Schler et al., 2006), Gaudian articles (Stamatatos, 2013) and IMDb reviews Seroussi et al. (2011) whose features are illustrated in Table 1. For the research papers dataset, each entry includes the title, abstract and introduction, and all of them are computer science papers gathered from arXiv[2], along with their ground-truth authorship information. We evaluate the capability of LLMs to accomplish this task under various setups, directly prompting both commercial platforms[3] such as GPT-4 (Achiam et al., 2023), GPT-3.5 (Ye et al., 2023), Claude-3.5 (Anthropic, 2024) and Kimi[4], as well as open-source models like Qwen (Bai et al., 2023) and Baichuan (Yang et al., 2023). Moreover, as the number of texts increases and exceeds the context window of LLMs, we propose a Retrieval-Augmented Generation (RAG)-based (Lewis et al., 2020; Gao et al., 2023; Ding et al., 2024b; Zhao et al., 2024) attribution method to address situations where the models cannot accommodate all the information even with context extension (Jin et al., 2024; Ding et al., 2024a; Zhang et al., 2024). The contributions of AIDBench are summarized as follows.

- AIDBench presents a novel dataset of research papers specifically designed to evaluate the authorship identification capabilities of LLMs in academic writing.

- Using AIDBench, we conduct extensive experiments of authorship identification with mainstream LLMs. These experiments are particularly performed under stringent conditions without any author profile information. Our empirical findings demonstrate that the potential of LLMs as tools for privacy breaches is evident and cannot be ignored.

---

[2]https://arxiv.org

[3]We have not tested all commercial models due to availability and cost constraints.

[4]https://kimi.moonshot.cn

- Additionally, we propose a Retrieval-Augmented Generation (RAG)-based methodological pipeline for evaluating the authorship identification capabilities of LLMs for large-scale authorship identification.

## 2 AIDBENCH DETAILS

In this section, we provide a detailed description of AIDBench. We begin by presenting an overview of the benchmark, outlining the pipeline for authorship identification using large language models (LLMs). We then describe the datasets used in our benchmark, the evaluation tasks conducted, and the metrics employed for assessment.

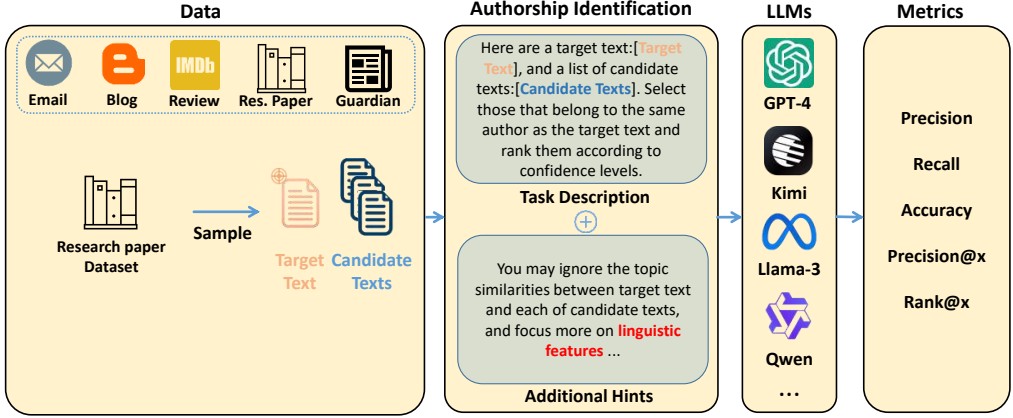

Figure 1: Overview of AIDBench.

### 2.1 OUTLINE OF AUTHORSHIP IDENTIFICATION WITH LLMS

Figure 1 provides an overview of our proposed AIDBench framework. We begin by selecting a dataset for evaluation, such as the Research Paper dataset. From this dataset, we sample a subset of texts from several authors, randomly selecting one as the *Target Text* and designating the remaining texts as candidates.

These texts are then incorporated into an authorship identification prompt, which is presented to the LLMs. The models generate responses indicating which candidate texts are more likely authored by the same individual as the *Target Text*. We repeat this process multiple times to obtain average performance metrics for the task. Finally, we employ metrics such as precision, recall, and rank to provide a clear and intuitive assessment of the LLMs' capabilities.

Table 1: Summary of datasets used in AIDBench

| Dataset | # of Authors | # of Texts | Text Length | Description | New or existing |
|---|---|---|---|---|---|
| Research Paper | 1,500 | 24,095 | 4,000∼7,000 | **A** collection of research papers on ArXiv with tag CS.LG from 2019 to 2024, with each author at least 10 papers after removing repeating entries. | **New** |
| Enron Email | 174 | 8,700 | 197 | **The** dataset is obtained from the original Enron email dataset by removing sender/receiver information and short emails. | (Klimt & Yang, 2004) |
| Blog | 1,500 | 15,000 | 116 | **The** dataset is sampled from the Blog Authorship Corpus with 19,320 posts from bloggers on blogger.com in August 2004. | (Schler et al., 2006) |
| IMDb Review | 62 | 3,100 | 340 | **The** dataset is filtered from the IMDb62 dataset by removing reviews with less than 10 words. | (Seroussi et al., 2011) |
| Guardian | 13 | 650 | 1060 | **The** texts are published in The Guardian daily newspaper, mainly opinion articles. Articles are from 13 authors across 5 topics. | (Stamatatos, 2013) |

## 2.2 DATASETS

AIDBench comprises five datasets: *Research Paper*, *Enron Email*, *Blog*, *IMDb Review*, and *Guardian*. In this subsection, we provide detailed descriptions of each dataset.

**Research Paper**. This newly collected dataset consists of research papers posted on arXiv under the CS.LG tag (the field of machine learning in the computer science domain) from 2019 to 2024. After removing duplicate entries and authors with fewer than ten papers, the dataset includes 24,095 papers from 1,500 authors, ensuring that each author has at least ten papers. We use this dataset to investigate their potential privacy risks when using LLMs for identifying authorship of academic writing.

**Enron Email**. The Enron email dataset contains approximately 500,000 emails generated by employees of the Enron Corporation. For our benchmark, similar to Huang et al. (2024a), we removed sender and receiver information and discarded short emails. Ultimately, we retained 174 authors, each with 50 emails.

**Blog**. The Blog Authorship Corpus (Schler et al., 2006) comprises the collected posts of 19,320 bloggers gathered from Blogger.com[5] in August 2004. We selected 1,500 authors from the dataset, each with 10 blog posts. The content of this dataset is closely related to daily life, providing linguistic characteristics, writing styles, word usage habits, and special characters that can be used to infer an author's identity.

**IMDb Review**. The IMDb review data are selected from the IMDb62 dataset (Seroussi et al., 2011). After filtering out reviews with fewer than 10 words, we randomly retained 50 reviews for each of the 62 authors in our dataset.

**Guardian**. The Guardian corpus dataset (Stamatatos, 2013) is designed to explore cross-genre and cross-topic authorship attribution. The corpus comprises texts published in *The Guardian* daily newspaper, mainly opinion articles (comments). It includes articles from 13 authors across five topics—politics, society, UK, world, and books—and retains 50 articles per author.

## 2.3 EVALUATION TASKS

As illustrated in Figure 1, we outline the evaluation tasks. Each task involves a query text and a set of candidate texts, with the objective of identifying which candidates are most likely authored by the same individual as the query text. Based on the number of candidate texts, we introduce the following specific tasks.

**One-to-one identification.** Commonly known as authorship verification, this task involves a single candidate text and aims to determine whether the text pair is authored by the same person, forming a binary classification problem. In our evaluation, we prompt the LLM with: "Here are a pair of texts: [Text Pair]. Determine if they belong to the same author.". More effective prompts or additional information can be provided to the LLM to achieve more accurate and reasonable responses. For instance, Hung et al. (2023) employs a Chain-of-Thought prompt and a series of intermediate reasoning steps to significantly enhance the LLMs' authorship verification ability. Similarly, as shown in Huang et al. (2024a) and Hung et al. (2023), instructing LLMs to analyze texts based on writing style, linguistic characteristics, and word usage habits, rather than content and topics, can further improve authorship verification.

Due to the subjective nature of the *one-to-one identification* task and its heavy reliance on the intrinsic judgment of LLMs, it is natural to consider a scenario involving two candidate texts requiring a comparative decision. The objective here is to determine which candidate text is more likely authored by the same individual as the query text; we refer to this as the **one-to-two identification** task. Due to space limitations, we defer the task design, metrics, and experimental results for the *one-to-two identification* task to Appendix B.

**One-to-many identification.** In this setup, we provide a number of candidate texts and ask LLMs to determine texts that are most likely authored by the same person as the text in the query. This is different from the usual closed-set authorship attribution task (Huang et al., 2024a), where multiple authors and their writings are provided in the context of LLMs, and the task is to attribute a target

---

[5] https://www.blogger.com

text to one of these authors. We do not provide any authorship information to LLMs. Instead in our experiments, we randomly sample a number of authors and put all of their writings into a set. Then one specific text is chosen at random as the target text while the rest from the set form the candidate texts. The task is to ask LLMs to identify the texts in the candidate set that are mostly likely authored by the same person as the target text, ranking the results by confidence scores.

## 2.4 RAG-BASED ONE-TO-MANY IDENTIFICATION PIPELINE

The performance of *one-to-many authorship identification* tasks heavily depends on the ability of LLMs to handle long contexts. Current commercial LLMs support relatively large context windows; for example, GPT-4-Turbo can process contexts up to 128,000 tokens, and Kimi supports up to 2 million tokens. In contrast, open-source models like Llama-3-8B-Instruct[6] support context windows of only up to 8,000 tokens. Moreover, the authorship identification task relies heavily on the LLMs' capacity for high-level comprehension of the context, which becomes increasingly challenging as the length of the context grows.

To address this limitation, we propose a simple Retrieval-Augmented Generation (RAG)-based method, as illustrated in Figure 2. In this approach, we first use pre-trained embedding models like sentence-transformers to encode both the target text and the candidate texts. We then calculate the similarities between the target text and each candidate, selecting only the top $k$ texts with the highest similarity scores. These top $k$ texts are provided to the LLM, thereby reducing the context length and enabling more effective processing.

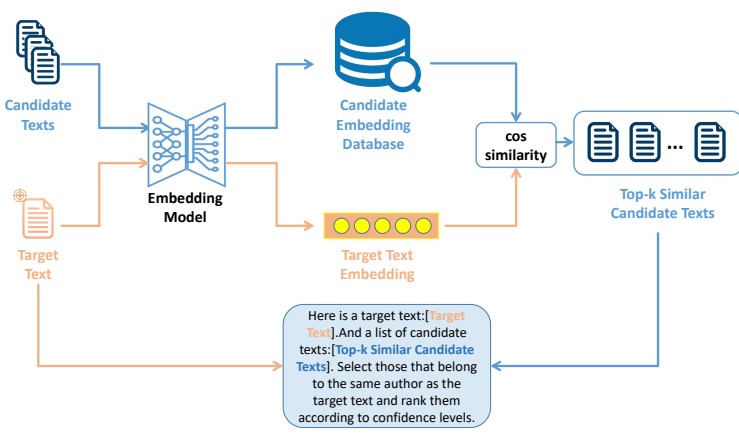

Figure 2: The RAG-based method for long context.

The motivation behind this method is that although embedding models tend to focus more on the semantic meanings of texts than on their linguistic characteristics, we can still use the similarities they provide to filter out texts that are significantly dissimilar to the target text. This approach ensures that the total length of the candidate texts remains within the context windows of LLMs.

## 2.5 EVALUATION METRICS

For the *one-to-one identification* task, LLMs are required to classify whether a pair of texts were written by the same author by answering "True" or "False". While accuracy can serve as an evaluation metric, it may not fully capture performance due to potential class imbalances in the dataset, for instance, when there is a large number of negative pairs compared to positive pairs, or vice versa. Therefore, we adopt *Precision, Recall* as our evaluation criteria to more comprehensively assess the performance of LLMs under various setups. Specifically, Precision and Recall are defined as follows:

$$\text{Precision} = \frac{\text{\#True Positive Pairs}}{\text{\#Predicted Positive Pairs}}; \qquad \text{Recall} = \frac{\text{\#True Positive Pairs}}{\text{\#Positive Pairs}}. \qquad (1)$$

---

[6] https://huggingface.co/meta-llama/Meta-Llama-3-8B-Instruct

For the *one-to-many identification* task, we employ multiple evaluation metrics to capture the problem's complex nature. In our experiments, we instruct the models to rank candidate texts based on confidence scores and repeat the experiments using multiple target texts. We utilize metrics such as *Rank@1* and *Rank@5* in our benchmark, where *Rank@x* indicates that at least one text authored by the same individual as the target text is found within the top $x$ ranked texts. Additionally, since we expect the LLMs to select as many correct candidate texts as possible, we use *Precision@x* as a metric, where *Precision@x* represents the proportion of correct predictions among the top $x$ selected texts.

## 3 EVALUATION RESULTS

In this section, we perform authorship identification tasks with the above introduced datasets on popular LLMs, e.g., GPT-3.5-turbo, GPT-4-turbo, Claude-3.5-sonnet, Kimi, Baichuan2-Turbo-192k, and Qwen1.5-72B-Chat[7]. In the following, we present two tasks (one-to-one and one-to-many identification tasks) on two representative datasets (Guardian and Research Paper), respectively. Other setups and results are deferred to Appendix.

### 3.1 ONE-TO-ONE IDENTIFICATION

As previously mentioned, the Guardian dataset categorizes text data by both author and topic. Leveraging this, we design a more challenging scenario to evaluate the one-to-one identification capabilities of LLMs. Specifically, we randomly select pairs of texts as positive examples where each pair is authored by the same individual but covers different topics. For negative examples, we sample text pairs that share the same topic but are written by different authors. We believe this setup provides a robust evaluation of LLM capabilities and offers deeper insights into their potential. For datasets other than the Guardian, we simply sample text pairs where the positive examples consist of texts by the same author, and the negative examples consist of texts by different authors.

We conduct experiments using three different random seeds, recording the precision and recall, and reporting the mean and standard deviation of these metrics. The total number of examples is set to 200, and we perform experiments with positive-to-negative example ratios of 150/50, 100/100, and 50/150, respectively.

Table 2: Evaluation of one-to-one identification on Guardian(%), where #/# denotes Pos./Neg. ratio.

| | 150/50 | | 100/100 | | 50/150 | |
| --- | --- | --- | --- | --- | --- | --- |
| | **Precision** | **Recall** | **Precision** | **Recall** | **Precision** | **Recall** |
| GPT-3.5-turbo$^\diamond$ | $69.0_{\pm 18.0}$ | $4.5_{\pm 2.1}$ | $47.8_{\pm 19.7}$ | $5.3_{\pm 2.1}$ | $21.5_{\pm 13.2}$ | $6.7_{\pm 5.0}$ |
| GPT-4-turbo$^\diamond$ | $90.2_{\pm 0.1}$ | $48.7_{\pm 2.9}$ | $74.2_{\pm 3.7}$ | $51.3_{\pm 1.2}$ | $45.5_{\pm 3.2}$ | $55.3_{\pm 5.0}$ |
| Kimi$^\diamond$ | $78.1_{\pm 3.3}$ | $22.3_{\pm 4.1}$ | $48.3_{\pm 13.8}$ | $41.3_{\pm 5.5}$ | $16.9_{\pm 6.0}$ | $14.7_{\pm 6.1}$ |
| Baichuan2-turbo-192k$^\diamond$ | $75.0_{\pm 0.9}$ | $95.1_{\pm 0.8}$ | $51.0_{\pm 0.8}$ | $97.3_{\pm 1.5}$ | $25.0_{\pm 0.5}$ | $96.0_{\pm 3.5}$ |
| Qwen1.5-72B-Chat$^\diamond$ | $82.0_{\pm 3.3}$ | $46.3_{\pm 5.8}$ | $56.4_{\pm 2.9}$ | $45.3_{\pm 6.5}$ | $32.2_{\pm 3.0}$ | $46.0_{\pm 3.5}$ |

As shown in Table 2, our evaluation results on the Guardian dataset indicate that precision decreases as the proportion of positive examples decreases. This suggests that all the models we evaluated tend to predict many negative examples as positive. In scenarios where there are fewer positive examples than negative ones, the performance of LLMs on the Guardian dataset is even worse than random guessing. The Baichuan model performs the worst, with precision close to the proportion of positive examples in the dataset and recall near 1, indicating that it almost considers all text pairs to be written by the same author. In fact, under the 50/150 setting, the LLMs encounter significant failures in evaluation on many other datasets (results are shown in Appendix D.1). These evaluation results differ significantly from the conclusions of prior works such as (Huang et al., 2024a) and (Hung et al., 2023). However, we still find that GPT-4-turbo exhibits very good precision on all datasets other than the Guardian, which we attribute to its superior reasoning capabilities compared to other models. This indicates that as LLMs continue to develop, they have the potential to become even more powerful tools for de-anonymization.

---

[7] https://huggingface.co/Qwen/Qwen1.5-72B-Chat

We believe that previous evaluation methods are not sufficiently rigorous and propose the following two points: First, the evaluation of LLMs' one-to-one identification capability should be conducted under cross-topic conditions, for which the Guardian dataset provides an excellent benchmark. Second, the proportion of negative examples in the dataset should be varied to assess whether LLMs incorrectly classify many negative examples as positive.

| General prompt | Topic-ignored prompt |
| --- | --- |
| Here is an abstract for a paper I want to attribute.
  attributed paper:{*attributed paper*}
  Below is a list containing the abstracts of multiple papers.
  Papers List:{*paper list*}
  Please try to identify and choose {*choose num*} papers in the list that most likely belong to the same author as the paper I want to attribute.
  Do not include as one of the options the self of the paper I want to attribute!
  Just output the ID of the paper you picked from the list directly, such as '1. paper ID: number'!
  Additionally, please sort the output based on confidence level, with the most likely papers appearing first. | Here is an abstract for a paper I want to attribute.
  attributed paper:{*attributed paper*}
  Below is a list containing the abstracts of multiple papers.
  Papers List:{*paper list*}
  Please try to identify and choose {*choose num*} papers in the list that most likely belong to the same author as the paper I want to attribute, ignoring differences in topic and content, focusing on linguistic features such as phrasal verbs, modal verbs, punctuation, rare words, affixes, numbers, humor, sarcasm, typographical errors, and spelling mistakes, and try to analyze the writing styles of the abstracts in the list and the abstract of the paper I want to attribute.
  Do not include as one of the options the self of the paper I want to attribute!
  Just output the ID of the paper you picked from the list directly, such as '1. paper ID: number'!
  Additionally, please sort the output based on confidence level, with the most likely papers appearing first. |

Figure 3: General prompt and Topic-ignored prompt

### 3.2 ONE-TO-MANY IDENTIFICATION

In this section, we evaluate the one-to-many identification capabilities of LLMs with the Research Paper dataset. We consider three scenarios: **2 Authors** with 20 papers; **5 Authors** with 50 papers; and **10 Authors** with 100 papers.

Table 3: Random guess on the **Research paper** dataset (%).

|  | 2 Authors | | | 5 Authors | | |
| --- | --- | --- | --- | --- | --- | --- |
|  | **Rank@1** | **Rank@3** | **Rank@5** | **Rank@1** | **Rank@3** | **Rank@5** |
| Random guess$^\diamond$ | 47.4 | 87.6 | 97.8 | 18.4 | 46.4 | 65.5 |
|  | **Precision@1** | **Precision@3** | **Precision@5** | **Precision@1** | **Precision@3** | **Precision@5** |
| Random guess$^\diamond$ | 47.4 | 47.4 | 47.4 | 18.4 | 18.4 | 18.4 |

To evaluate each scenario, we construct subsets from the Research Paper dataset. Using the two-authors scenario as an example, we randomly select two authors from a pool of 1,500 authors. This selection process is repeated ten times to create ten distinct groups, each containing two authors. For each group, we randomly choose a target paper, which remains the same across all models evaluated within that group to ensure fairness. Within each group, we repeat the evaluation process three times to compute the mean rank and precision metrics.

Subsequently, we prompt the models to perform one-to-many authorship identification for this scenario. Finally, we calculate the mean and standard deviation of each metric across the ten groups, providing an overall assessment of the models' performance.

Tables 4 and 5 present the one-to-many identification results obtained by prompting the models with topic-ignored prompts and general prompts, respectively, across different scenarios. Additionally, for the typical long text scenario involving ten authors, Figure 4 illustrates the comparison of one-to-many identification performance between different models and between the two types of prompts for the same model.

Table 4: Evaluation of model one-to-many identification capability on the **Research Paper** dataset using ***topic-ignored*** prompts in the right of Fig. 3 (%).

| | 2 Authors | | | 5 Authors | | |
|---|---|---|---|---|---|---|
| | **Rank@1** | **Rank@3** | **Rank@5** | **Rank@1** | **Rank@3** | **Rank@5** |
| GPT-3.5-turbo$^\diamond$ | $70.0_{\pm24.6}$ | $86.7_{\pm17.2}$ | $86.7_{\pm17.2}$ | $10.0_{\pm16.1}$ | $46.7_{\pm23.3}$ | $70.0_{\pm24.6}$ |
| GPT-4-turbo$^\diamond$ | $83.3_{\pm32.4}$ | $93.3_{\pm14.1}$ | $100.0_{\pm0.0}$ | $53.3_{\pm23.3}$ | $80.0_{\pm17.2}$ | $86.7_{\pm17.2}$ |
| Kimi$^\diamond$ | $83.3_{\pm17.5}$ | $83.3_{\pm17.5}$ | $86.7_{\pm17.2}$ | $43.3_{\pm27.4}$ | $53.3_{\pm28.1}$ | $60.0_{\pm21.1}$ |
| Baichuan2-turbo-192k$^\diamond$ | $60.0_{\pm30.6}$ | $70.0_{\pm24.6}$ | $70.0_{\pm24.6}$ | $36.7_{\pm24.6}$ | $46.7_{\pm23.3}$ | $63.3_{\pm22.5}$ |
| Qwen1.5-72B-Chat$^\diamond$ | $56.7_{\pm22.5}$ | $76.7_{\pm22.5}$ | $86.7_{\pm17.2}$ | $13.3_{\pm17.2}$ | $23.3_{\pm22.5}$ | $33.3_{\pm27.2}$ |
| | **Precision@1** | **Precision@3** | **Precision@5** | **Precision@1** | **Precision@3** | **Precision@5** |
| GPT-3.5-turbo$^\diamond$ | $70.0_{\pm24.6}$ | $68.9_{\pm21.5}$ | $60.0_{\pm14.7}$ | $10.0_{\pm16.1}$ | $17.8_{\pm10.7}$ | $20.0_{\pm9.4}$ |
| GPT-4-turbo$^\diamond$ | $83.3_{\pm32.4}$ | $73.3_{\pm21.7}$ | $70.7_{\pm10.5}$ | $53.3_{\pm23.3}$ | $43.3_{\pm9.7}$ | $33.3_{\pm7.0}$ |
| Kimi$^\diamond$ | $83.3_{\pm17.5}$ | $70.2_{\pm21.6}$ | $64.7_{\pm23.1}$ | $43.3_{\pm27.4}$ | $30.1_{\pm15.8}$ | $20.0_{\pm10.4}$ |
| Baichuan2-turbo-192k$^\diamond$ | $60.0_{\pm30.6}$ | $60.0_{\pm26.8}$ | $58.0_{\pm24.2}$ | $36.7_{\pm24.6}$ | $33.3_{\pm20.9}$ | $30.0_{\pm18.1}$ |
| Qwen1.5-72B-Chat$^\diamond$ | $56.7_{\pm22.5}$ | $51.1_{\pm19.7}$ | $46.7_{\pm12.2}$ | $13.3_{\pm17.2}$ | $8.9_{\pm8.8}$ | $8.0_{\pm6.9}$ |

Table 5: Evaluation of model one-to-many identification capability on the **Research paper** dataset using ***general*** prompts in the left of Fig. 3 (%).

| | 2 Authors | | | 5 Authors | | |
|---|---|---|---|---|---|---|
| | **Rank@1** | **Rank@3** | **Rank@5** | **Rank@1** | **Rank@3** | **Rank@5** |
| GPT-3.5-turbo$^\diamond$ | $56.7_{\pm35.3}$ | $86.7_{\pm17.2}$ | $96.7_{\pm10.5}$ | $30.0_{\pm24.6}$ | $60.0_{\pm26.3}$ | $73.3_{\pm21.1}$ |
| GPT-4-turbo$^\diamond$ | $80.0_{\pm23.3}$ | $93.3_{\pm14.1}$ | $93.3_{\pm14.1}$ | $70.0_{\pm29.2}$ | $86.7_{\pm17.2}$ | $90.0_{\pm16.1}$ |
| Kimi$^\diamond$ | $86.7_{\pm17.2}$ | $86.7_{\pm17.2}$ | $93.3_{\pm14.1}$ | $50.0_{\pm17.6}$ | $63.3_{\pm10.5}$ | $73.3_{\pm14.1}$ |
| Baichuan2-turbo-192k$^\diamond$ | $70.0_{\pm29.2}$ | $80.0_{\pm23.3}$ | $80.0_{\pm23.3}$ | $46.7_{\pm23.3}$ | $60.0_{\pm30.6}$ | $66.7_{\pm31.4}$ |
| Qwen1.5-72B-Chat$^\diamond$ | $70.0_{\pm29.2}$ | $76.7_{\pm31.6}$ | $83.3_{\pm23.6}$ | $10.0_{\pm16.1}$ | $23.3_{\pm27.4}$ | $26.7_{\pm26.3}$ |
| | **Precision@1** | **Precision@3** | **Precision@5** | **Precision@1** | **Precision@3** | **Precision@5** |
| GPT-3.5-turbo$^\diamond$ | $56.7_{\pm35.3}$ | $55.6_{\pm18.9}$ | $55.3_{\pm12.6}$ | $30.0_{\pm24.6}$ | $25.6_{\pm16.6}$ | $20.0_{\pm10.4}$ |
| GPT-4-turbo$^\diamond$ | $80.0_{\pm23.3}$ | $72.2_{\pm15.9}$ | $70.0_{\pm15.8}$ | $70.0_{\pm29.2}$ | $64.4_{\pm20.8}$ | $53.3_{\pm18.6}$ |
| Kimi$^\diamond$ | $86.7_{\pm17.2}$ | $76.7_{\pm25.9}$ | $72.7_{\pm24.4}$ | $50.0_{\pm17.6}$ | $41.1_{\pm18.2}$ | $34.0_{\pm14.2}$ |
| Baichuan2-turbo-192k$^\diamond$ | $70.0_{\pm29.2}$ | $68.9_{\pm22.1}$ | $64.7_{\pm20.4}$ | $46.7_{\pm23.3}$ | $43.3_{\pm27.4}$ | $43.3_{\pm25.0}$ |
| Qwen1.5-72B-Chat$^\diamond$ | $70.0_{\pm29.2}$ | $50.0_{\pm23.6}$ | $45.3_{\pm14.0}$ | $10.0_{\pm16.1}$ | $10.0_{\pm14.3}$ | $7.3_{\pm9.1}$ |

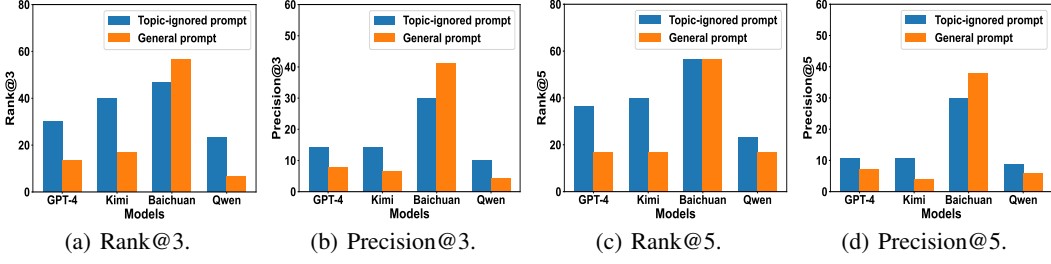

(a) Rank@3.    (b) Precision@3.    (c) Rank@5.    (d) Precision@5.

Figure 4: Evaluation and Comparison in **10 Authors** scenario. The context length of **10 Authors** already exceeds the limit of GPT-3.5-turbo.

The results indicate that in short-context scenarios, such as the one with two authors, GPT-4-turbo and Kimi demonstrate a clear advantage and significantly outperform the random guessing baseline shown in Figure 3. As the context length increases, for example, in the scenario with five authors, GPT-4-turbo continues to maintain its lead among the models. However, some of Kimi's performance metrics, such as *rank@5* and *precision@3*, begin to be surpassed by Baichuan2-Turbo-192k. In the ten-authors scenario shown in Figure 4, which tests the models' context window and long-context processing capabilities, GPT-3-turbo is excluded from the competition due to its limited context window size. Interestingly, in this scenario, Baichuan2-Turbo-192k significantly outperforms the other models, in stark contrast to its performance in the two-author and five-author scenarios. Notably, Baichuan2-Turbo-192k performs better with the general prompt than with the topic-ignored prompt, suggesting that incorporating the paper's topic enhances authorship identification for this model.

Table 6: Evaluation of model one-to-many identification capability on the **First author subset of Research paper** dataset using ***topic-ignored*** prompts (%).

| | 5 Authors | | | 10 Authors | | |
| | Rank@1 | Rank@3 | Rank@5 | Rank@1 | Rank@3 | Rank@5 |
|---|---|---|---|---|---|---|
| GPT-4-turbo$^\diamond$ | $66.67_{\pm 0.00}$ | $90.00_{\pm 16.10}$ | $93.33_{\pm 14.05}$ | $66.67_{\pm 0.00}$ | $83.33_{\pm 17.57}$ | $90.00_{\pm 16.10}$ |
| Qwen1.5-72B-Chat$^\diamond$ | $63.33_{\pm 10.54}$ | $83.33_{\pm 23.57}$ | $86.67_{\pm 23.31}$ | $66.67_{\pm 0.00}$ | $80.00_{\pm 17.21}$ | $86.67_{\pm 17.21}$ |
| Claude-3.5-sonnet$^\diamond$ | $90.00_{\pm 22.50}$ | $96.67_{\pm 10.54}$ | $96.67_{\pm 10.54}$ | $63.33_{\pm 42.89}$ | $73.33_{\pm 34.43}$ | $73.33_{\pm 34.43}$ |
| | Precision@1 | Precision@3 | Precision@5 | Precision@1 | Precision@3 | Precision@5 |
| GPT-4-turbo$^\diamond$ | $66.67_{\pm 0.00}$ | $72.22_{\pm 9.44}$ | $71.33_{\pm 9.96}$ | $66.67_{\pm 0.00}$ | $63.33_{\pm 10.54}$ | $61.33_{\pm 10.80}$ |
| Qwen1.5-72B-Chat$^\diamond$ | $63.33_{\pm 10.54}$ | $66.67_{\pm 13.86}$ | $64.00_{\pm 16.69}$ | $66.67_{\pm 0.00}$ | $55.56_{\pm 11.71}$ | $52.00_{\pm 11.24}$ |
| Claude-3.5-sonnet$^\diamond$ | $90.00_{\pm 22.50}$ | $91.11_{\pm 14.63}$ | $79.33_{\pm 13.13}$ | $63.33_{\pm 42.89}$ | $56.67_{\pm 32.48}$ | $48.67_{\pm 24.95}$ |

Table 7: Evaluation of model one-to-many identification capability on the **First author subset of Research paper** dataset using ***general*** prompts (%).

| | 5 Authors | | | 10 Authors | | |
| | Rank@1 | Rank@3 | Rank@5 | Rank@1 | Rank@3 | Rank@5 |
|---|---|---|---|---|---|---|
| GPT-4-turbo$^\diamond$ | $66.67_{\pm 0.00}$ | $93.33_{\pm 14.05}$ | $93.33_{\pm 14.05}$ | $66.67_{\pm 0.00}$ | $90.00_{\pm 16.10}$ | $93.33_{\pm 14.05}$ |
| Qwen1.5-72B-Chat$^\diamond$ | $66.67_{\pm 0.00}$ | $83.33_{\pm 17.57}$ | $86.67_{\pm 17.21}$ | $66.67_{\pm 0.00}$ | $83.33_{\pm 17.57}$ | $83.33_{\pm 17.57}$ |
| Claude-3.5-sonnet$^\diamond$ | $86.67_{\pm 17.21}$ | $93.33_{\pm 14.05}$ | $93.33_{\pm 14.05}$ | $73.33_{\pm 21.08}$ | $73.33_{\pm 21.08}$ | $76.67_{\pm 22.50}$ |
| | Precision@1 | Precision@3 | Precision@5 | Precision@1 | Precision@3 | Precision@5 |
| GPT-4-turbo$^\diamond$ | $66.67_{\pm 0.00}$ | $71.11_{\pm 7.77}$ | $71.33_{\pm 7.73}$ | $66.67_{\pm 0.00}$ | $68.89_{\pm 8.76}$ | $65.33_{\pm 9.32}$ |
| Qwen1.5-72B-Chat$^\diamond$ | $66.67_{\pm 0.00}$ | $67.78_{\pm 9.73}$ | $67.33_{\pm 12.35}$ | $66.67_{\pm 0.00}$ | $63.33_{\pm 12.88}$ | $56.00_{\pm 10.52}$ |
| Claude-3.5-sonnet$^\diamond$ | $86.67_{\pm 17.21}$ | $85.56_{\pm 17.41}$ | $74.00_{\pm 13.86}$ | $73.33_{\pm 21.08}$ | $70.00_{\pm 23.45}$ | $56.67_{\pm 26.90}$ |

**Evaluation on the first-author subset.** The involvement of multiple authors in a single paper can introduce confounding factors that may lead to inaccurate assessments of authorship identification capabilities. To mitigate this issue, we curated a subset of papers where each author is represented solely by their first-author publications, ensuring that each has more than five such papers. We posit that the first author is most likely to lead and contribute significantly to the writing, allowing for a more accurate evaluation.

Using this subset, we conduct experiments with three representative models from both open-source and closed-source categories. Our results show that among these models, Claude-3.5-sonnet demonstrate remarkable authorship identification capabilities. Additionally, GPT-4-turbo and Qwen1.5-72B-Chat exhibit significantly improved performance on the first-author subset compared to previous evaluations. These findings suggest that the models encountered less interference when identifying authorship in the first-author subset, further highlighting the potential risk of LLMs in revealing the identities of anonymous authors.

Table 8: Evaluation of searching-based baseline in 2 Authors and 5 Authors scenarios(%).

| | Rank@1 | Rank@3 | Rank@5 | Precision@1 | Precision@3 | Precision@5 |
|---|---|---|---|---|---|---|
| **2 Authors** | $70.0_{\pm 33.1}$ | $93.3_{\pm 14.1}$ | $96.7_{\pm 10.5}$ | $70.0_{\pm 33.1}$ | $61.1_{\pm 20.5}$ | $55.3_{\pm 16.6}$ |
| **5 Authors** | $46.0_{\pm 26.7}$ | $80.0_{\pm 13.3}$ | $90.0_{\pm 14.1}$ | $46.0_{\pm 26.7}$ | $41.3_{\pm 14.3}$ | $36.8_{\pm 9.9}$ |

**Comparison with searching-based baseline.** Moreover, in addition to the evaluation of several large language models, we further evaluate and compare the searching-based baseline with our proposed pipeline in Section 2.4. For the evaluation of the searching-based baseline, we first use an embedding model[8] to convert the text into embedding. Then, we calculate the cosine similarity between the embedding of the attributed paper and the embeddings of the other papers. The top 10 results with the highest similarity are extracted to compute the rank and precision.

---

[8] https://huggingface.co/sentence-transformers/all-mpnet-base-v2

Table 9: Evaluation and Comparison of searching-based baseline and RAG-based one-to-many identification pipeline(%).

| | 50 papers filter 10 papers | | | 200 papers filter 50 papers | | |
|---|---|---|---|---|---|---|
| | Rank@1 | Rank@3 | Rank@5 | Rank@1 | Rank@3 | Rank@5 |
| Searching♠ | $46.0_{\pm 26.7}$ | $80.0_{\pm 13.3}$ | $90.0_{\pm 14.1}$ | $10.0_{\pm 14.1}$ | $18.0_{\pm 11.4}$ | $26.0_{\pm 13.5}$ |
| RAG+Qwen♡ | $54.0_{\pm 25.0}$ | $78.0_{\pm 11.4}$ | $94.0_{\pm 9.7}$ | $14.0_{\pm 13.5}$ | $22.3_{\pm 14.7}$ | $26.1_{\pm 18.3}$ |
| RAG+GPT-3.5-turbo♡ | $40.0_{\pm 16.3}$ | $72.0_{\pm 16.9}$ | $86.0_{\pm 13.5}$ | $2.0_{\pm 6.3}$ | $12.0_{\pm 16.8}$ | $16.0_{\pm 20.1}$ |
| RAG+GPT-4-turbo♡ | $54.0_{\pm 23.0}$ | $76.0_{\pm 18.0}$ | $84.0_{\pm 13.0}$ | $40.0_{\pm 31.3}$ | $52.0_{\pm 25.3}$ | $56.0_{\pm 22.7}$ |
| RAG+Kimi♡ | $64.0_{\pm 20.7}$ | $76.0_{\pm 15.8}$ | $86.0_{\pm 13.5}$ | $50.0_{\pm 28.7}$ | $60.0_{\pm 28.3}$ | $64.0_{\pm 26.3}$ |
| RAG+Baichuan♡ | $48.0_{\pm 14.0}$ | $74.0_{\pm 21.2}$ | $82.0_{\pm 19.9}$ | $8.0_{\pm 19.3}$ | $12.0_{\pm 21.5}$ | $14.0_{\pm 21.2}$ |
| | Precision@1 | Precision@3 | Precision@5 | Precision@1 | Precision@3 | Precision@5 |
| Searching♠ | $46.0_{\pm 26.7}$ | $41.3_{\pm 14.3}$ | $36.8_{\pm 9.9}$ | $10.0_{\pm 14.1}$ | $8.7_{\pm 5.5}$ | $7.6_{\pm 4.4}$ |
| RAG+Qwen♡ | $54.0_{\pm 25.0}$ | $41.3_{\pm 8.2}$ | $35.2_{\pm 4.1}$ | $14.0_{\pm 13.5}$ | $11.3_{\pm 8.3}$ | $7.6_{\pm 5.8}$ |
| RAG+GPT-3.5-turbo♡ | $40.0_{\pm 16.3}$ | $40.7_{\pm 8.6}$ | $36.4_{\pm 6.1}$ | $2.0_{\pm 6.3}$ | $4.0_{\pm 5.6}$ | $4.0_{\pm 4.6}$ |
| RAG+GPT-4-turbo♡ | $54.0_{\pm 23.0}$ | $44.7_{\pm 15.7}$ | $37.6_{\pm 9.7}$ | $40.0_{\pm 31.3}$ | $25.3_{\pm 16.6}$ | $20.0_{\pm 11.0}$ |
| RAG+Kimi♡ | $64.0_{\pm 20.7}$ | $50.0_{\pm 9.6}$ | $39.2_{\pm 9.2}$ | $50.0_{\pm 28.7}$ | $38.7_{\pm 24.3}$ | $28.4_{\pm 17.2}$ |
| RAG+Baichuan♡ | $48.0_{\pm 14.0}$ | $47.3_{\pm 15.5}$ | $36.4_{\pm 10.7}$ | $8.0_{\pm 19.3}$ | $8.0_{\pm 16.9}$ | $5.2_{\pm 10.0}$ |

Table 8 presents the evaluation results of the searching-based method in the 2 Authors and 5 Authors scenarios. Due to the straightforward semantic similarity calculation, the performance of this method declines sharply in longer text scenarios. The results of the comparison between the searching-based method and our pipeline method are displayed in Table 9. In the RAG-based one-to-many identification pipeline, we consistently use the topic-ignored prompt shown in the right of Fig. 3. We find that our proposed pipeline, which combines RAG with certain LLMs such as GPT-4-turbo and Kimi, performs better in one-to-many identification. Additionally, the "50 papers filter 10 paper" scenario corresponds to the 5 Authors scenario in Table 4. Comparing the data from these two parts reveals that the RAG-based method significantly improves both rank and precision.

# 4  CONCLUSION

In this work, we propose a benchmark to evaluate the authorship identification capabilities of large language models (LLMs). Specifically, we have collected a new dataset, referred to as the "Research Paper" dataset, to identify authorship based solely on the content of academic papers. By combining this with various existing public datasets, we have curated a comprehensive benchmark dataset. We evaluate the authorship identification task using multiple models and services, revealing potential privacy risks in real-world anonymous systems when employing LLMs. To enable authorship identification of long texts, we have devised a pipeline utilizing Retrieval-Augmented Generation (RAG). This approach allows us to tailor the evaluation of LLMs with different contextual capacities to adapt to the task. Further discussion on limitations and future directions is deferred to Appendix C.

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

# Appendix

## A    RELATED WORK

Our benchmark is related to the longstanding authorship attribution task (Stamatatos, 2009), which is initially formulated as supervised authorship identification and recently addressed with LLMs.

### A.1    SUPERVISED AUTHORSHIP IDENTIFICATION

Early work uses text statistics such as n-grams, topic, or syntax as features (Koppel et al., 2007; Sharma et al., 2018; Sundararajan & Woodard, 2018; Seroussi et al., 2014), and then leverage SVM, random forest or topic models to conduct authorship verification. Some work uses deep learning models like RNNs and CNNs to extract more complex features for authorship verification (Bagnall, 2015; Benzebouchi et al., 2018; Ruder et al., 2016). Pretrained models such as BERT are also used to get text embeddings (Huertas-Tato et al., 2022; Barlas & Stamatatos, 2020a; Fabien et al., 2020; Rivera-Soto et al., 2021) and one can verify the authorship of two texts by measuring the similarity of their embeddings.

### A.2    AUTHORSHIP IDENTIFICATION BY LLMS

Due to the abilities of LLMs understanding texts and human instructions, researchers naturally explore using LLMs for authorship identification. Hung et al. (2023) prompt LLMs with step-by-step stylometric explanations for authorship verification. Huang et al. (2024a) design various prompts with different levels of guidance and ask LLMs to conduct authorship verification and attribution in an end-to-end way. The guidance includes ignoring topics, ignoring content, and focusing on linguistic features. Nonetheless, they did not experiment with hard scenarios such as the same author with different topics, or different authors with the same topic.

Our paper extends previous work in several key aspects. We focus on an authorship identification task that mimics the anonymous paper review system by identifying the most likely texts from the same author. Additionally, our experiments cover a wider range of scenarios and demonstrate the behaviors of LLMs in various axes, including different context lengths and challenging scenarios.

## B    RESULTS OF ONE-TO-TWO IDENTIFICATION TASK

**One-to-two identification.** The previous task heavily relies on the intrinsic judgment of LLMs due to its subjective nature. Therefore, we consider a scenario with two candidate texts, requiring a comparative decision. The objective is to determine which candidate text is more likely authored by the same person as the query text. Specifically, we prompt the LLM with, "Here are two candidate texts: [candidate texts] and a target text [text in query]. Determine which candidate text is more likely to be authored by the same person as the target text". Similar to the previous task, the prompts can be augmented with additional information.

We choose GPT-3.5-turbo, GPT-4-turbo, Kimi and Baichuan2-turbo-192k to perform one-to-two identification on 4 datasets: Guardian, Enron Email, Blog and IMDb Review. We use three different random seeds to repeat our experiments and choose accuracy as the evaluation metric.

In this task, each data example consists of three texts: besides the target text, there are two candidate texts. One of the candidate texts is by the same author as the target text, while the other is not. We sample 200 data examples for our experiments. It is worth noticing that for the Guardian dataset, we ensure that the target text and the candidate text belonging to the same author as the target text are from different topics, while the other candidate text has the same topic as the candidate text. The evaluation results are shown in table 10.

The results in Table 10 show that GPT-4-turbo still demonstrates a significant advantage in one-to-two identification tasks, the accuracy reaches an average of 70.0% on Guardian, 80.7% on Email, 80.0% on Blog, 75.8% on IMDb. This indicates that although texts from different authors may be similar in topic, GPT-4-turbo can make reasonable comparisons based on the linguistic characteristics of the texts, thereby identifying the candidate text that truly belongs to the same author as the

Table 10: Accuracy(%) of one-to-two identification.

|  | Guardian | Email | Blog | IMDb |
|---|---|---|---|---|
| GPT-3.5-turbo$^\diamond$ | $54.8_{\pm 1.9}$ | $72.2_{\pm 1.2}$ | $74.2_{\pm 1.5}$ | $69.0_{\pm 3.6}$ |
| GPT-4-turbo$^\diamond$ | $70.0_{\pm 4.0}$ | $80.7_{\pm 1.4}$ | $80.0_{\pm 1.7}$ | $75.8_{\pm 3.8}$ |
| Kimi$^\diamond$ | $58.3_{\pm 4.0}$ | $66.0_{\pm 2.2}$ | $65.0_{\pm 2.3}$ | $63.5_{\pm 4.4}$ |
| Baichuan2-turbo-192k$^\diamond$ | $53.7_{\pm 4.3}$ | $63.5_{\pm 1.3}$ | $65.5_{\pm 1.8}$ | $56.3_{\pm 5.6}$ |

target text, which demonstrates a considerable privacy threat. Though all four models that we evaluate show effectiveness in this task, there is a significant difference in accuracy between different models, which suggests that there is a considerable variance in the abilities of different models to analyze and compare texts and understand instructions.

## C LIMITATION AND FUTURE DIRECTIONS

The AIDBench offers a benchmark for assessing LLMs' authorship identification capabilities in scenarios closer to real-world anonymous review systems, highlighting significant privacy risks associated with anonymous systems. In this section, we address the potential limitations of AIDBench and outline future directions.

Firstly, it is worth noting that due to cost constraints, we evaluated only a limited number of models. Additionally, large commercial models are continually emerging and evolving rapidly, such as the Gemini (Team et al., 2023), Claude 3 (Anthropic, 2024), Llama (Touvron et al., 2023a;b; Meta, 2024), Mistral families (Jiang et al., 2023), and many others (Zeng et al., 2022; Zhang et al., 2022a). Therefore, future exploration and evaluation of these models are warranted.

Another aspect to consider is that research papers often involve multiple authors, each contributing differently to the paper's writing. It would be intriguing to curate more nuanced subsets of research papers based on author positions, such as a first-author subset. This approach could provide insights into how author position affects the authorship identification problem in research papers. Additionally, in our experiments, we only utilize the abstract information of a paper. However, in the future, incorporating more detailed paper information could lead to a more accurate evaluation.

Furthermore, in the evaluation process, we employed only two types of prompts: the general prompt and the topic-ignored prompt. Recognizing the significant impact of prompt quality on model responses, it is essential for a more comprehensive exploration of LLMs' authorship identification capabilities that we develop a more nuanced set of prompts for future evaluation and analysis.

Finally, the RAG-based approach we introduced enables LLMs with limited context windows to conduct authorship identification. To ensure the success of authorship identification, it's necessary to limit the candidate list obtained through RAG screening to a specific quantity. Additionally, we often encounter individual texts with considerable lengths. Therefore, the future direction of our work involves exploring methods to decompose long texts into shorter segments, indirectly facilitating comparisons between the original texts by comparing these shorter segments. One approach is to segment the long text into several shorter segments. In the one-to-two identification task, LLMs can compare pairs of these shorter segments, assigning scores during each comparison. Ultimately, scores are weighted and summed based on the length of each segment.

## D SUPPLEMENTAL EXPERIMENTAL RESULTS

### D.1 ONE-TO-ONE RESULTS

This subsection provides the supplementary evaluation results of one-to-one identification in table 11-13. We evaluate 5 models: GPT-3.5-turbo, GPT-4-turbo, Kimi, Baichuan2-turbo-192k, Qwen1.5-72B-Chat on 3 datasets: Blog, Enron Email, IMDb Review.

From these results, we can have the following findings. Firstly, precisions decrease as the proportion of positive examples in the dataset decreases, which is consistent with the conclusions and proposals

Table 11: Evaluation of one-to-one identification on Blog(%).

| | 150/50 | | 100/100 | | 50/150 | |
|---|---|---|---|---|---|---|
| | **Precision** | **Recall** | **Precision** | **Recall** | **Precision** | **Recall** |
| GPT-3.5-turbo$^\diamond$ | $78.7_{\pm9.0}$ | $10.4_{\pm2.8}$ | $62.7_{\pm9.5}$ | $14.7_{\pm1.2}$ | $32.6_{\pm13.3}$ | $10.7_{\pm3.1}$ |
| GPT-4-turbo$^\diamond$ | $96.1_{\pm1.3}$ | $49.1_{\pm1.0}$ | $89.8_{\pm5.7}$ | $48.0_{\pm2.6}$ | $79.0_{\pm5.0}$ | $49.3_{\pm1.2}$ |
| Kimi$^\diamond$ | $78.9_{\pm1.6}$ | $68.0_{\pm3.3}$ | $57.1_{\pm1.9}$ | $69.3_{\pm3.5}$ | $31.3_{\pm2.4}$ | $71.3_{\pm4.2}$ |
| Baichuan2-turbo-192k$^\diamond$ | $78.0_{\pm0.6}$ | $77.8_{\pm3.3}$ | $54.8_{\pm0.6}$ | $83.3_{\pm3.8}$ | $30.3_{\pm0.5}$ | $83.3_{\pm1.2}$ |
| Qwen1.5-72B-Chat$^\diamond$ | $84.9_{\pm2.2}$ | $63.8_{\pm1.7}$ | $69.9_{\pm1.8}$ | $63.3_{\pm0.6}$ | $44.3_{\pm2.5}$ | $63.3_{\pm1.2}$ |

Table 12: Evaluation of one-to-one identification on Enron Email(%).

| | 150/50 | | 100/100 | | 50/150 | |
|---|---|---|---|---|---|---|
| | **Precision** | **Recall** | **Precision** | **Recall** | **Precision** | **Recall** |
| GPT-3.5-turbo$^\diamond$ | $77.1_{\pm6.7}$ | $14.9_{\pm4.0}$ | $52.2_{\pm7.7}$ | $15.0_{\pm3.5}$ | $28.2_{\pm13.0}$ | $12.7_{\pm8.1}$ |
| GPT-4-turbo$^\diamond$ | $99.4_{\pm0.1}$ | $38.4_{\pm4.0}$ | $99.3_{\pm1.2}$ | $37.3_{\pm7.8}$ | $98.2_{\pm3.0}$ | $34.7_{\pm6.1}$ |
| Kimi$^\diamond$ | $95.3_{\pm2.8}$ | $49.8_{\pm3.4}$ | $87.4_{\pm0.5}$ | $51.0_{\pm4.0}$ | $68.2_{\pm5.7}$ | $46.7_{\pm3.1}$ |
| Baichuan2-turbo-192k$^\diamond$ | $81.4_{\pm0.5}$ | $74.7_{\pm2.9}$ | $58.9_{\pm2.7}$ | $75.3_{\pm6.8}$ | $27.9_{\pm0.7}$ | $70.0_{\pm5.3}$ |
| Qwen1.5-32B-Chat$^\diamond$ | $97.0_{\pm3.0}$ | $45.8_{\pm5.4}$ | $90.4_{\pm1.2}$ | $47.3_{\pm6.8}$ | $78.4_{\pm10.6}$ | $44.7_{\pm3.1}$ |

Table 13: Evaluation of one-to-one identification on IMDb Review(%).

| | 150/50 | | 100/100 | | 50/150 | |
|---|---|---|---|---|---|---|
| | **Precision** | **Recall** | **Precision** | **Recall** | **Precision** | **Recall** |
| GPT-3.5-turbo$^\diamond$ | $88.6_{\pm12.7}$ | $5.8_{\pm1.5}$ | $70.0_{\pm11.2}$ | $5.7_{\pm1.5}$ | $29.7_{\pm16.5}$ | $7.3_{\pm4.6}$ |
| GPT-4-turbo$^\diamond$ | $93.6_{\pm2.3}$ | $58.4_{\pm2.8}$ | $88.5_{\pm1.5}$ | $56.0_{\pm3.6}$ | $74.9_{\pm2.5}$ | $59.3_{\pm3.1}$ |
| Kimi$^\diamond$ | $85.3_{\pm0.5}$ | $43.8_{\pm9.7}$ | $78.6_{\pm6.6}$ | $49.0_{\pm4.6}$ | $52.7_{\pm7.5}$ | $52.7_{\pm5.0}$ |
| Baichuan2-turbo-192k$^\diamond$ | $75.4_{\pm1.6}$ | $66.7_{\pm4.1}$ | $53.0_{\pm2.7}$ | $68.7_{\pm4.5}$ | $28.6_{\pm1.9}$ | $70.0_{\pm6.9}$ |
| Qwen1.5-32B-Chat$^\diamond$ | $100.0_{\pm0.0}$ | $9.1_{\pm1.6}$ | $95.8_{\pm7.2}$ | $10.7_{\pm3.5}$ | $100.0_{\pm0.0}$ | $13.3_{\pm4.2}$ |

given in the main text. Secondly, in many cases, the recall values are relatively low, which indicates that even without controlling for topics as in the Guardian dataset, there are still some challenging examples that make it difficult for LLMs to distinguish. Besides, the same model exhibits different performances under different datasets. This demonstrates the necessity of AIDBench providing multiple datasets for evaluation. We can also find that GPT-4-turbo consistently performs relatively well across various scenarios. This fact leads us to hypothesize that the capability for authorship identification is not static, it evolves and strengthens as LLMs develop.

## D.2 ONE-TO-MANY RESULTS

In addition to evaluating the one-to-many identification capabilities of multiple models on research paper datasets, we also conducted experiments on blog and email datasets.

Tables 14 and 15, along with Figure 5, present detailed experimental data on the blog dataset. GPT-4-turbo continues to perform well, demonstrating strong long-text processing capabilities and instruction-following abilities. Unlike the evaluation results on research papers in Section 3.2 of the main text, Qwen1.5-72B-Chat performs overall better than Kimi and Baichuan2-turbo-192k on the blog dataset. We hypothesize two reasons for this phenomenon. First, each blog in the dataset has a distinct personality and is more closely tied to the author, without the extensive influence of research fields and similar thematic articles as seen in the research paper dataset. Second, the shorter length of each blog reduces the reliance on the model's long-text processing capabilities.

As shown, Tables 16 and 17, as well as Figure 6, present the evaluation results on the email dataset. It is noteworthy that across the three scenarios in the email dataset, namely 2 Authors, 5 Authors, and

Table 14: Evaluation of model one-to-many identification capability on the **Blog** dataset using ***topic-ignored*** prompt in the right of Fig. 3 (%).

| | 2 Authors | | | 5 Authors | | |
|---|---|---|---|---|---|---|
| | **Rank@1** | **Rank@3** | **Rank@5** | **Rank@1** | **Rank@3** | **Rank@5** |
| GPT-3.5-turbo$^\diamond$ | $60.0_{\pm14.1}$ | $70.0_{\pm18.9}$ | $83.3_{\pm23.6}$ | $36.7_{\pm33.2}$ | $50.0_{\pm28.3}$ | $60.0_{\pm26.3}$ |
| GPT-4-turbo$^\diamond$ | $86.7_{\pm23.3}$ | $90.0_{\pm16.1}$ | $96.7_{\pm10.5}$ | $86.7_{\pm17.2}$ | $90.0_{\pm16.1}$ | $90.0_{\pm16.1}$ |
| Kimi$^\diamond$ | $73.3_{\pm14.1}$ | $73.3_{\pm14.1}$ | $76.7_{\pm16.1}$ | $63.3_{\pm29.2}$ | $66.7_{\pm31.4}$ | $70.0_{\pm24.6}$ |
| Baichuan2-turbo-192k$^\diamond$ | $73.3_{\pm26.3}$ | $80.0_{\pm28.1}$ | $86.7_{\pm23.3}$ | $43.3_{\pm31.6}$ | $50.0_{\pm28.3}$ | $53.3_{\pm28.1}$ |
| Qwen1.5-72B-Chat$^\diamond$ | $66.7_{\pm0.0}$ | $93.3_{\pm14.1}$ | $93.3_{\pm14.1}$ | $66.7_{\pm0.0}$ | $83.3_{\pm17.6}$ | $86.7_{\pm17.2}$ |
| | **Precision@1** | **Precision@3** | **Precision@5** | **Precision@1** | **Precision@3** | **Precision@5** |
| GPT-3.5-turbo$^\diamond$ | $60.0_{\pm14.1}$ | $56.7_{\pm19.2}$ | $59.3_{\pm20.0}$ | $36.7_{\pm33.2}$ | $33.3_{\pm20.3}$ | $30.7_{\pm16.1}$ |
| GPT-4-turbo$^\diamond$ | $86.7_{\pm23.3}$ | $84.4_{\pm23.5}$ | $82.7_{\pm22.5}$ | $86.7_{\pm17.2}$ | $76.7_{\pm22.5}$ | $67.3_{\pm21.4}$ |
| Kimi$^\diamond$ | $73.3_{\pm14.1}$ | $68.9_{\pm18.0}$ | $66.0_{\pm18.2}$ | $63.3_{\pm29.2}$ | $51.1_{\pm32.0}$ | $46.0_{\pm30.2}$ |
| Baichuan2-turbo-192k$^\diamond$ | $73.3_{\pm26.3}$ | $72.2_{\pm27.8}$ | $67.3_{\pm26.8}$ | $43.3_{\pm31.6}$ | $34.4_{\pm25.9}$ | $25.3_{\pm21.5}$ |
| Qwen1.5-72B-Chat$^\diamond$ | $66.7_{\pm0.0}$ | $68.9_{\pm8.8}$ | $69.3_{\pm9.0}$ | $66.7_{\pm0.0}$ | $52.2_{\pm7.5}$ | $47.3_{\pm6.6}$ |

Table 15: Evaluation of model one-to-many identification capability on the **Blog** dataset using ***general*** prompt in the left of Fig. 3 (%).

| | 2 Authors | | | 5 Authors | | |
|---|---|---|---|---|---|---|
| | **Rank@1** | **Rank@3** | **Rank@5** | **Rank@1** | **Rank@3** | **Rank@5** |
| GPT-3.5-turbo$^\diamond$ | $60.0_{\pm30.6}$ | $73.3_{\pm21.1}$ | $83.3_{\pm17.6}$ | $56.7_{\pm22.5}$ | $73.3_{\pm21.1}$ | $83.3_{\pm17.6}$ |
| GPT-4-turbo$^\diamond$ | $86.7_{\pm23.3}$ | $86.7_{\pm23.3}$ | $86.7_{\pm23.3}$ | $80.0_{\pm28.1}$ | $86.7_{\pm23.3}$ | $93.3_{\pm14.1}$ |
| Kimi$^\diamond$ | $76.7_{\pm22.5}$ | $76.7_{\pm22.5}$ | $80.0_{\pm23.3}$ | $56.7_{\pm35.3}$ | $60.0_{\pm37.8}$ | $66.7_{\pm31.4}$ |
| Baichuan2-turbo-192k$^\diamond$ | $66.7_{\pm27.2}$ | $73.3_{\pm26.3}$ | $80.0_{\pm23.3}$ | $40.0_{\pm30.6}$ | $50.0_{\pm23.6}$ | $56.7_{\pm22.5}$ |
| Qwen1.5-72B-Chat$^\diamond$ | $66.7_{\pm0.0}$ | $93.3_{\pm14.1}$ | $93.3_{\pm14.1}$ | $66.7_{\pm0.0}$ | $86.7_{\pm17.2}$ | $86.7_{\pm17.2}$ |
| | **Precision@1** | **Precision@3** | **Precision@5** | **Precision@1** | **Precision@3** | **Precision@5** |
| GPT-3.5-turbo$^\diamond$ | $60.0_{\pm30.6}$ | $58.9_{\pm22.3}$ | $64.0_{\pm15.8}$ | $56.7_{\pm22.5}$ | $44.4_{\pm18.9}$ | $43.3_{\pm18.4}$ |
| GPT-4-turbo$^\diamond$ | $86.7_{\pm23.3}$ | $84.4_{\pm22.4}$ | $84.0_{\pm23.1}$ | $80.0_{\pm28.1}$ | $73.3_{\pm28.3}$ | $66.7_{\pm23.9}$ |
| Kimi$^\diamond$ | $76.7_{\pm22.5}$ | $74.4_{\pm20.3}$ | $73.3_{\pm18.1}$ | $56.7_{\pm35.3}$ | $47.8_{\pm35.2}$ | $42.0_{\pm30.2}$ |
| Baichuan2-turbo-192k$^\diamond$ | $66.7_{\pm27.2}$ | $63.3_{\pm30.6}$ | $59.3_{\pm29.2}$ | $40.0_{\pm30.6}$ | $36.7_{\pm24.0}$ | $38.0_{\pm23.9}$ |
| Qwen1.5-72B-Chat$^\diamond$ | $66.7_{\pm0.0}$ | $73.3_{\pm7.8}$ | $70.0_{\pm8.5}$ | $66.7_{\pm0.0}$ | $60.0_{\pm9.4}$ | $56.7_{\pm10.1}$ |

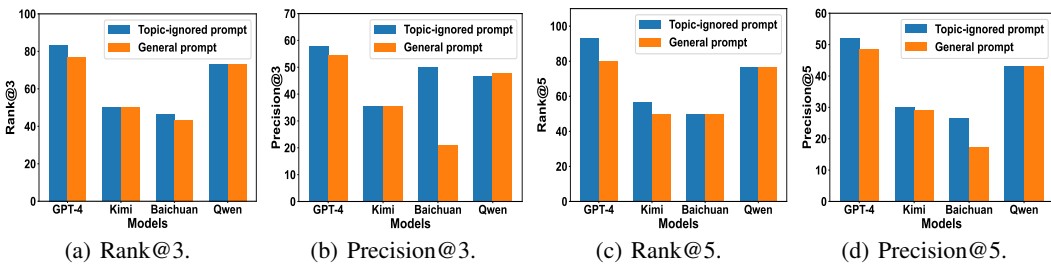

(a) Rank@3.  (b) Precision@3.  (c) Rank@5.  (d) Precision@5.

Figure 5: Evaluation and Comparison in 10 Authors scenario of **Blog** dataset.

10 Authors, the comparative trends among the models are relatively consistent, yet the performance differences between the models are not substantial. Upon examining the email dataset, we found that some emails contain identity information of the senders and recipients, which could be one of the reasons for the observed phenomenon.

# E  ABLATION EXPERIMENT

We further conducted ablation experiments to thoroughly investigate the impact of each factor within the prompt on the model's ability to identify the authorship. As shown in Fig. 7, we meticulously

Table 16: Evaluation of model one-to-many identification capability on the **Enron Email** dataset using *topic-ignored* prompt in the right of Fig. 3 (%).

| | 2 Authors | | | 5 Authors | | |
| --- | --- | --- | --- | --- | --- | --- |
| | **Rank@1** | **Rank@3** | **Rank@5** | **Rank@1** | **Rank@3** | **Rank@5** |
| GPT-3.5-turbo$^\diamond$ | $56.7_{\pm38.7}$ | $70.0_{\pm29.2}$ | $80.0_{\pm28.1}$ | $66.7_{\pm0.0}$ | $76.7_{\pm16.1}$ | $83.3_{\pm17.6}$ |
| GPT-4-turbo$^\diamond$ | $66.7_{\pm0.0}$ | $93.3_{\pm14.1}$ | $96.7_{\pm10.5}$ | $66.7_{\pm0.0}$ | $90.0_{\pm16.1}$ | $90.0_{\pm16.1}$ |
| Kimi$^\diamond$ | $66.7_{\pm0.0}$ | $86.7_{\pm17.2}$ | $86.7_{\pm17.2}$ | $66.7_{\pm0.0}$ | $83.3_{\pm17.6}$ | $83.3_{\pm17.6}$ |
| Baichuan2-turbo-192k$^\diamond$ | $66.7_{\pm0.0}$ | $86.7_{\pm17.2}$ | $90.0_{\pm16.1}$ | $66.7_{\pm0.0}$ | $86.7_{\pm17.2}$ | $90.0_{\pm16.1}$ |
| Qwen1.5-72B-Chat$^\diamond$ | $66.7_{\pm0.0}$ | $93.3_{\pm14.1}$ | $100.0_{\pm0.0}$ | $66.7_{\pm0.0}$ | $73.3_{\pm14.1}$ | $80.0_{\pm17.2}$ |
| | **Precision@1** | **Precision@3** | **Precision@5** | **Precision@1** | **Precision@3** | **Precision@5** |
| GPT-3.5-turbo$^\diamond$ | $56.7_{\pm38.7}$ | $55.6_{\pm31.4}$ | $55.3_{\pm27.0}$ | $66.7_{\pm0.0}$ | $51.1_{\pm5.7}$ | $48.0_{\pm5.3}$ |
| GPT-4-turbo$^\diamond$ | $66.7_{\pm0.0}$ | $75.6_{\pm4.7}$ | $77.3_{\pm4.7}$ | $66.7_{\pm0.0}$ | $64.4_{\pm12.6}$ | $60.7_{\pm12.7}$ |
| Kimi$^\diamond$ | $66.7_{\pm0.0}$ | $71.1_{\pm7.8}$ | $72.0_{\pm9.3}$ | $66.7_{\pm0.0}$ | $58.9_{\pm10.5}$ | $55.3_{\pm12.2}$ |
| Baichuan2-turbo-192k$^\diamond$ | $66.7_{\pm0.0}$ | $67.8_{\pm9.7}$ | $68.7_{\pm10.0}$ | $66.7_{\pm0.0}$ | $53.3_{\pm7.0}$ | $48.0_{\pm6.9}$ |
| Qwen1.5-72B-Chat$^\diamond$ | $66.7_{\pm0.0}$ | $68.9_{\pm8.8}$ | $66.0_{\pm6.6}$ | $66.7_{\pm0.0}$ | $53.3_{\pm8.8}$ | $49.3_{\pm9.0}$ |

Table 17: Evaluation of model one-to-many identification capability on the **Enron Email** dataset using *general* prompt in the left of Fig. 3 (%).

| | 2 Authors | | | 5 Authors | | |
| --- | --- | --- | --- | --- | --- | --- |
| | **Rank@1** | **Rank@3** | **Rank@5** | **Rank@1** | **Rank@3** | **Rank@5** |
| GPT-3.5-turbo$^\diamond$ | $63.3_{\pm10.5}$ | $80.0_{\pm17.2}$ | $83.3_{\pm17.6}$ | $63.3_{\pm10.5}$ | $76.7_{\pm16.1}$ | $83.3_{\pm17.6}$ |
| GPT-4-turbo$^\diamond$ | $66.7_{\pm0.0}$ | $100.0_{\pm0.0}$ | $100.0_{\pm0.0}$ | $66.7_{\pm0.0}$ | $86.7_{\pm17.2}$ | $86.7_{\pm17.2}$ |
| Kimi$^\diamond$ | $66.7_{\pm0.0}$ | $86.7_{\pm17.2}$ | $86.7_{\pm17.2}$ | $66.7_{\pm0.0}$ | $83.3_{\pm17.6}$ | $83.3_{\pm17.6}$ |
| Baichuan2-turbo-192k$^\diamond$ | $66.7_{\pm0.0}$ | $90.0_{\pm16.1}$ | $90.0_{\pm16.1}$ | $66.7_{\pm0.0}$ | $83.3_{\pm17.6}$ | $86.7_{\pm17.2}$ |
| Qwen1.5-72B-Chat$^\diamond$ | $66.7_{\pm0.0}$ | $93.3_{\pm14.1}$ | $93.3_{\pm14.1}$ | $66.7_{\pm0.0}$ | $73.3_{\pm14.1}$ | $73.3_{\pm14.1}$ |
| | **Precision@1** | **Precision@3** | **Precision@5** | **Precision@1** | **Precision@3** | **Precision@5** |
| GPT-3.5-turbo$^\diamond$ | $63.3_{\pm10.5}$ | $65.6_{\pm11.0}$ | $64.0_{\pm11.4}$ | $63.3_{\pm10.5}$ | $48.9_{\pm7.8}$ | $47.3_{\pm10.2}$ |
| GPT-4-turbo$^\diamond$ | $66.7_{\pm0.0}$ | $76.7_{\pm3.5}$ | $78.7_{\pm4.2}$ | $66.7_{\pm0.0}$ | $60.0_{\pm10.7}$ | $57.3_{\pm11.4}$ |
| Kimi$^\diamond$ | $66.7_{\pm0.0}$ | $70.0_{\pm7.5}$ | $69.3_{\pm7.8}$ | $66.7_{\pm0.0}$ | $56.7_{\pm9.7}$ | $51.3_{\pm9.5}$ |
| Baichuan2-turbo-192k$^\diamond$ | $66.7_{\pm0.0}$ | $67.8_{\pm6.3}$ | $68.7_{\pm7.7}$ | $66.7_{\pm0.0}$ | $55.6_{\pm9.1}$ | $53.3_{\pm9.9}$ |
| Qwen1.5-72B-Chat$^\diamond$ | $66.7_{\pm0.0}$ | $68.9_{\pm7.0}$ | $65.3_{\pm6.1}$ | $66.7_{\pm0.0}$ | $48.9_{\pm5.7}$ | $45.3_{\pm6.1}$ |

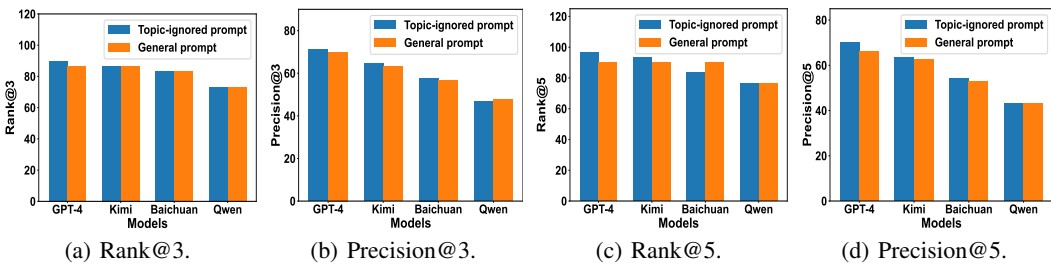

(a) Rank@3.     (b) Precision@3.     (c) Rank@5.     (d) Precision@5.

Figure 6: Evaluation and Comparison in 10 Authors scenario of **email** dataset.

studied the influence of linguistic features, writing styles, topics, and content on the ability of LLMs to identify authorship within prompts.

**Topic-ignored prompt**

Here is an abstract for a paper I want to attribute.
  attributed paper:{*attributed paper*}
  Below is a list containing the abstracts of multiple papers.
  Papers List:{*paper list*}
  Please try to identify and choose {choose num} papers in the list that most likely belong to the same author as the paper I want to attribute, ignoring differences in topic and content, focusing on linguistic features such as phrasal verbs, modal verbs, punctuation, rare words, affixes, numbers, humor, sarcasm, typographical errors, and spelling mistakes, and try to analyze the writing styles of the abstracts in the list and the abstract of the paper I want to attribute.
  Do not include as one of the options the self of the paper I want to attribute!
  Just output the ID of the paper you picked from the list directly, such as '1. paper ID: number'!
  Additionally, please sort the output based on confidence level, with the most likely papers appearing first.

**w/o linguistic features**

Here is an abstract for a paper I want to attribute.
  attributed paper:{*attributed paper*}
  Below is a list containing the abstracts of multiple papers.
  Papers List:{*paper list*}
  Please try to identify and choose {choose num} papers in the list that most likely belong to the same author as the paper I want to attribute, ignoring differences in topic and content, ~~focusing on linguistic features such as phrasal verbs, modal verbs, punctuation, rare words, affixes, numbers, humor, sarcasm, typographical errors, and spelling mistakes~~, and try to analyze the writing styles of the abstracts in the list and the abstract of the paper I want to attribute.
  Do not include as one of the options the self of the paper I want to attribute!
  Just output the ID of the paper you picked from the list directly, such as '1. paper ID: number'!
  Additionally, please sort the output based on confidence level, with the most likely papers appearing first.

**w/o writing styles**

Here is an abstract for a paper I want to attribute.
  attributed paper:{*attributed paper*}
  Below is a list containing the abstracts of multiple papers.
  Papers List:{*paper list*}
  Please try to identify and choose {choose num} papers in the list that most likely belong to the same author as the paper I want to attribute, ignoring differences in topic and content, focusing on linguistic features such as phrasal verbs, modal verbs, punctuation, rare words, affixes, numbers, humor, sarcasm, typographical errors, and spelling mistakes, ~~and try to analyze the writing styles~~ of the abstracts in the list and the abstract of the paper I want to attribute.
  Do not include as one of the options the self of the paper I want to attribute!
  Just output the ID of the paper you picked from the list directly, such as '1. paper ID: number'!
  Additionally, please sort the output based on confidence level, with the most likely papers appearing first.

**w/o topic and content**

Here is an abstract for a paper I want to attribute.
  attributed paper:{*attributed paper*}
  Below is a list containing the abstracts of multiple papers.
  Papers List:{*paper list*}
  Please try to identify and choose {choose num} papers in the list that most likely belong to the same author as the paper I want to attribute, ~~ignoring differences in topic and content,~~ focusing on linguistic features such as phrasal verbs, modal verbs, punctuation, rare words, affixes, numbers, humor, sarcasm, typographical errors, and spelling mistakes, and try to analyze the writing styles of the abstracts in the list and the abstract of the paper I want to attribute.
  Do not include as one of the options the self of the paper I want to attribute!
  Just output the ID of the paper you picked from the list directly, such as '1. paper ID: number'!
  Additionally, please sort the output based on confidence level, with the most likely papers appearing first.

Figure 7: Various prompts used in ablation experiments

Table 18: **GPT-4-turbo** ablation experiments on **Research Paper** dataset (%).

|  | 2 Authors | | | 5 Authors | | |
|---|---|---|---|---|---|---|
|  | Rank@1 | Rank@3 | Rank@5 | Rank@1 | Rank@3 | Rank@5 |
| w/o linguistic features$^\diamond$ | $66.7_{\pm13.9}$ | $83.3_{\pm10.5}$ | $100.0_{\pm0.0}$ | $56.7_{\pm27.4}$ | $70.0_{\pm24.6}$ | $80.0_{\pm23.3}$ |
| w/o writing styles$^\diamond$ | $70.0_{\pm29.2}$ | $100.0_{\pm0.0}$ | $100.0_{\pm0.0}$ | $63.3_{\pm24.6}$ | $83.3_{\pm23.6}$ | $83.3_{\pm23.6}$ |
| w/o topic and content$^\diamond$ | $76.7_{\pm22.5}$ | $83.3_{\pm17.6}$ | $90.0_{\pm16.1}$ | $63.3_{\pm26.3}$ | $73.3_{\pm26.3}$ | $86.7_{\pm23.3}$ |
| Original Prompt$^\spadesuit$ | $83.3_{\pm32.4}$ | $93.3_{\pm14.1}$ | $100.0_{\pm0.0}$ | $53.3_{\pm23.3}$ | $80.0_{\pm17.2}$ | $86.7_{\pm17.2}$ |
|  | Precision@1 | Precision@3 | Precision@5 | Precision@1 | Precision@3 | Precision@5 |
| w/o linguistic features$^\diamond$ | $66.7_{\pm13.9}$ | $74.4_{\pm18.9}$ | $68.7_{\pm17.8}$ | $56.7_{\pm27.4}$ | $42.2_{\pm18.0}$ | $40.0_{\pm17.5}$ |
| w/o writing styles$^\diamond$ | $70.0_{\pm29.2}$ | $70.0_{\pm18.2}$ | $64.7_{\pm17.5}$ | $63.3_{\pm24.6}$ | $56.7_{\pm19.2}$ | $42.0_{\pm16.9}$ |
| w/o topic and content$^\diamond$ | $76.7_{\pm22.5}$ | $66.7_{\pm13.9}$ | $61.3_{\pm14.7}$ | $63.3_{\pm18.9}$ | $52.2_{\pm18.9}$ | $46.0_{\pm13.9}$ |
| Original Prompt$^\spadesuit$ | $83.3_{\pm32.4}$ | $73.3_{\pm21.7}$ | $70.7_{\pm10.5}$ | $53.3_{\pm23.3}$ | $43.3_{\pm9.7}$ | $33.3_{\pm7.0}$ |

Table 19: **Qwen1.5-72B-Chat** ablation experiments on **Research Paper** dataset (%).

|  | 2 Authors | | | 5 Authors | | |
|---|---|---|---|---|---|---|
|  | Rank@1 | Rank@3 | Rank@5 | Rank@1 | Rank@3 | Rank@5 |
| w/o linguistic features$^\diamond$ | $70.0_{\pm29.2}$ | $83.3_{\pm23.6}$ | $83.3_{\pm23.6}$ | $23.3_{\pm16.1}$ | $40.0_{\pm26.3}$ | $43.3_{\pm22.5}$ |
| w/o writing styles$^\diamond$ | $63.3_{\pm18.9}$ | $66.7_{\pm22.2}$ | $73.3_{\pm21.1}$ | $26.7_{\pm26.3}$ | $30.0_{\pm24.6}$ | $33.3_{\pm27.2}$ |
| w/o topic and content$^\diamond$ | $80.0_{\pm17.2}$ | $90.0_{\pm16.1}$ | $96.7_{\pm10.5}$ | $23.3_{\pm22.5}$ | $30.0_{\pm18.9}$ | $36.7_{\pm18.9}$ |
| Original Prompt$^\spadesuit$ | $56.7_{\pm22.5}$ | $76.7_{\pm22.5}$ | $86.7_{\pm17.2}$ | $13.3_{\pm17.2}$ | $23.3_{\pm22.5}$ | $33.3_{\pm27.2}$ |
|  | Precision@1 | Precision@3 | Precision@5 | Precision@1 | Precision@3 | Precision@5 |
| w/o linguistic features$^\diamond$ | $70.0_{\pm29.2}$ | $50.0_{\pm10.8}$ | $42.7_{\pm11.0}$ | $23.3_{\pm16.1}$ | $21.1_{\pm15.2}$ | $13.3_{\pm8.3}$ |
| w/o writing styles$^\diamond$ | $63.3_{\pm18.9}$ | $51.1_{\pm25.8}$ | $46.0_{\pm22.3}$ | $26.7_{\pm26.3}$ | $15.6_{\pm15.0}$ | $10.7_{\pm10.5}$ |
| w/o topic and content$^\diamond$ | $80.0_{\pm17.2}$ | $66.7_{\pm15.7}$ | $52.0_{\pm11.7}$ | $23.3_{\pm22.5}$ | $14.4_{\pm11.8}$ | $12.0_{\pm8.8}$ |
| Original Prompt$^\spadesuit$ | $56.7_{\pm22.5}$ | $51.1_{\pm19.7}$ | $46.7_{\pm12.2}$ | $13.3_{\pm17.2}$ | $8.9_{\pm8.8}$ | $8.0_{\pm6.9}$ |

Table 20: **GPT-4-turbo** ablation experiments on **Enron Email** dataset (%).

| | 2 Authors | | | 5 Authors | | |
| --- | --- | --- | --- | --- | --- | --- |
| | **Rank@1** | **Rank@3** | **Rank@5** | **Rank@1** | **Rank@3** | **Rank@5** |
| w/o linguistic features$^{\diamond}$ | $100.0_{\pm 0.0}$ | $100.0_{\pm 0.0}$ | $100.0_{\pm 0.0}$ | $73.3_{\pm 30.6}$ | $80.0_{\pm 28.1}$ | $80.0_{\pm 28.1}$ |
| w/o writing styles$^{\diamond}$ | $86.7_{\pm 17.2}$ | $93.3_{\pm 14.1}$ | $96.7_{\pm 10.5}$ | $66.7_{\pm 27.2}$ | $76.7_{\pm 27.4}$ | $76.7_{\pm 27.4}$ |
| w/o topic and content$^{\diamond}$ | $83.3_{\pm 23.6}$ | $93.3_{\pm 14.1}$ | $93.3_{\pm 14.1}$ | $80.0_{\pm 28.1}$ | $80.0_{\pm 28.1}$ | $80.0_{\pm 28.1}$ |
| Original Prompt$^{\spadesuit}$ | $66.7_{\pm 0.0}$ | $93.3_{\pm 14.1}$ | $96.7_{\pm 10.5}$ | $66.7_{\pm 0.0}$ | $90.0_{\pm 16.1}$ | $90.0_{\pm 16.1}$ |
| | **Precision@1** | **Precision@3** | **Precision@5** | **Precision@1** | **Precision@3** | **Precision@5** |
| w/o linguistic features$^{\diamond}$ | $100.0_{\pm 0.0}$ | $94.4_{\pm 14.1}$ | $92.0_{\pm 18.8}$ | $73.3_{\pm 30.6}$ | $65.6_{\pm 29.4}$ | $62.0_{\pm 31.0}$ |
| w/o writing styles$^{\diamond}$ | $86.7_{\pm 17.2}$ | $87.8_{\pm 14.3}$ | $88.0_{\pm 15.3}$ | $66.7_{\pm 27.2}$ | $61.1_{\pm 28.8}$ | $54.7_{\pm 29.3}$ |
| w/o topic and content$^{\diamond}$ | $83.3_{\pm 23.6}$ | $83.3_{\pm 24.7}$ | $83.3_{\pm 22.9}$ | $80.0_{\pm 28.1}$ | $66.7_{\pm 27.7}$ | $60.0_{\pm 25.5}$ |
| Original Prompt$^{\spadesuit}$ | $66.7_{\pm 0.0}$ | $75.6_{\pm 4.7}$ | $77.3_{\pm 4.7}$ | $66.7_{\pm 0.0}$ | $64.4_{\pm 12.6}$ | $60.7_{\pm 12.7}$ |

Table 21: **Qwen1.5-72B-Chat** ablation experiments on **Enron Email** dataset (%).

| | 2 Authors | | | 5 Authors | | |
| --- | --- | --- | --- | --- | --- | --- |
| | **Rank@1** | **Rank@3** | **Rank@5** | **Rank@1** | **Rank@3** | **Rank@5** |
| w/o linguistic features$^{\diamond}$ | $70.0_{\pm 29.2}$ | $73.3_{\pm 30.6}$ | $80.0_{\pm 23.3}$ | $16.7_{\pm 23.6}$ | $33.3_{\pm 31.4}$ | $36.7_{\pm 29.2}$ |
| w/o writing styles$^{\diamond}$ | $80.0_{\pm 23.3}$ | $86.7_{\pm 23.3}$ | $93.3_{\pm 14.1}$ | $23.3_{\pm 22.5}$ | $40.0_{\pm 26.3}$ | $46.7_{\pm 23.3}$ |
| w/o topic and content$^{\diamond}$ | $80.0_{\pm 32.2}$ | $80.0_{\pm 32.2}$ | $86.7_{\pm 23.3}$ | $26.7_{\pm 26.3}$ | $33.3_{\pm 31.4}$ | $33.3_{\pm 31.4}$ |
| Original Prompt$^{\spadesuit}$ | $66.7_{\pm 0.0}$ | $93.3_{\pm 14.1}$ | $100.0_{\pm 0.0}$ | $66.7_{\pm 0.0}$ | $73.3_{\pm 14.1}$ | $80.0_{\pm 17.2}$ |
| | **Precision@1** | **Precision@3** | **Precision@5** | **Precision@1** | **Precision@3** | **Precision@5** |
| w/o linguistic features$^{\diamond}$ | $70.0_{\pm 29.2}$ | $64.4_{\pm 26.1}$ | $58.7_{\pm 21.0}$ | $16.7_{\pm 23.6}$ | $14.4_{\pm 13.9}$ | $9.3_{\pm 7.8}$ |
| w/o writing styles$^{\diamond}$ | $80.0_{\pm 23.3}$ | $77.8_{\pm 26.2}$ | $67.3_{\pm 21.9}$ | $23.3_{\pm 22.5}$ | $21.1_{\pm 16.1}$ | $18.7_{\pm 14.3}$ |
| w/o topic and content$^{\diamond}$ | $80.0_{\pm 32.2}$ | $68.9_{\pm 33.5}$ | $63.3_{\pm 24.2}$ | $26.7_{\pm 26.3}$ | $18.9_{\pm 22.3}$ | $13.3_{\pm 16.9}$ |
| Original Prompt$^{\spadesuit}$ | $66.7_{\pm 0.0}$ | $68.9_{\pm 8.8}$ | $66.0_{\pm 6.6}$ | $66.7_{\pm 0.0}$ | $53.3_{\pm 8.8}$ | $49.3_{\pm 9.0}$ |

Table 22: **GPT-4-turbo** ablation experiments on **Blog** dataset (%).

| | 2 Authors | | | 5 Authors | | |
| --- | --- | --- | --- | --- | --- | --- |
| | **Rank@1** | **Rank@3** | **Rank@5** | **Rank@1** | **Rank@3** | **Rank@5** |
| w/o linguistic features$^{\diamond}$ | $86.7_{\pm 23.3}$ | $96.7_{\pm 10.5}$ | $96.7_{\pm 10.5}$ | $80.0_{\pm 28.1}$ | $80.0_{\pm 28.1}$ | $83.3_{\pm 23.6}$ |
| w/o writing styles$^{\diamond}$ | $80.0_{\pm 23.3}$ | $86.7_{\pm 17.2}$ | $93.3_{\pm 14.1}$ | $70.0_{\pm 24.6}$ | $73.3_{\pm 21.1}$ | $83.3_{\pm 23.6}$ |
| w/o topic and content$^{\diamond}$ | $90.0_{\pm 16.1}$ | $90.0_{\pm 16.1}$ | $90.0_{\pm 16.1}$ | $60.0_{\pm 26.3}$ | $73.3_{\pm 26.3}$ | $83.3_{\pm 28.3}$ |
| Original Prompt$^{\spadesuit}$ | $86.7_{\pm 23.3}$ | $90.0_{\pm 16.1}$ | $96.7_{\pm 10.5}$ | $86.7_{\pm 17.2}$ | $90.0_{\pm 16.1}$ | $90.0_{\pm 16.1}$ |
| | **Precision@1** | **Precision@3** | **Precision@5** | **Precision@1** | **Precision@3** | **Precision@5** |
| w/o linguistic features$^{\diamond}$ | $86.7_{\pm 23.3}$ | $87.8_{\pm 13.3}$ | $84.0_{\pm 17.6}$ | $80.0_{\pm 28.1}$ | $66.7_{\pm 24.0}$ | $62.0_{\pm 24.2}$ |
| w/o writing styles$^{\diamond}$ | $80.0_{\pm 23.3}$ | $81.1_{\pm 21.0}$ | $80.0_{\pm 17.8}$ | $70.0_{\pm 24.6}$ | $60.0_{\pm 23.5}$ | $54.0_{\pm 23.6}$ |
| w/o topic and content$^{\diamond}$ | $90.0_{\pm 16.1}$ | $83.3_{\pm 20.5}$ | $81.3_{\pm 22.6}$ | $60.0_{\pm 26.3}$ | $56.7_{\pm 23.1}$ | $52.7_{\pm 28.2}$ |
| Original Prompt$^{\spadesuit}$ | $86.7_{\pm 23.3}$ | $84.4_{\pm 23.5}$ | $82.7_{\pm 22.5}$ | $86.7_{\pm 17.2}$ | $76.7_{\pm 22.5}$ | $67.3_{\pm 21.4}$ |

Table 23: **Qwen1.5-72B-Chat** ablation experiments on **Blog** dataset (%).

| | 2 Authors | | | 5 Authors | | |
| --- | --- | --- | --- | --- | --- | --- |
| | **Rank@1** | **Rank@3** | **Rank@5** | **Rank@1** | **Rank@3** | **Rank@5** |
| w/o linguistic features$^{\diamond}$ | $83.3_{\pm 23.6}$ | $83.3_{\pm 23.6}$ | $93.3_{\pm 14.1}$ | $26.7_{\pm 21.1}$ | $33.3_{\pm 22.2}$ | $36.7_{\pm 24.6}$ |
| w/o writing styles$^{\diamond}$ | $86.7_{\pm 17.2}$ | $90.0_{\pm 16.1}$ | $90.0_{\pm 16.1}$ | $40.0_{\pm 30.6}$ | $46.7_{\pm 32.2}$ | $46.7_{\pm 32.2}$ |
| w/o topic and content$^{\diamond}$ | $70.0_{\pm 18.9}$ | $83.3_{\pm 17.6}$ | $83.3_{\pm 17.6}$ | $50.0_{\pm 39.3}$ | $56.7_{\pm 31.6}$ | $56.7_{\pm 31.6}$ |
| Original Prompt$^{\spadesuit}$ | $66.7_{\pm 0.0}$ | $93.3_{\pm 14.1}$ | $93.3_{\pm 14.1}$ | $66.7_{\pm 0.0}$ | $83.3_{\pm 17.6}$ | $86.7_{\pm 17.2}$ |
| | **Precision@1** | **Precision@3** | **Precision@5** | **Precision@1** | **Precision@3** | **Precision@5** |
| w/o linguistic features$^{\diamond}$ | $83.3_{\pm 23.6}$ | $76.7_{\pm 25.9}$ | $72.7_{\pm 23.8}$ | $26.7_{\pm 21.1}$ | $16.7_{\pm 13.1}$ | $12.0_{\pm 10.8}$ |
| w/o writing styles$^{\diamond}$ | $86.7_{\pm 17.2}$ | $72.2_{\pm 21.8}$ | $60.7_{\pm 20.2}$ | $40.0_{\pm 30.6}$ | $25.6_{\pm 19.6}$ | $17.3_{\pm 15.5}$ |
| w/o topic and content$^{\diamond}$ | $70.0_{\pm 18.9}$ | $71.1_{\pm 15.9}$ | $61.3_{\pm 13.3}$ | $50.0_{\pm 39.3}$ | $34.4_{\pm 27.9}$ | $22.0_{\pm 16.6}$ |
| Original Prompt$^{\spadesuit}$ | $66.7_{\pm 0.0}$ | $68.9_{\pm 8.8}$ | $69.3_{\pm 9.0}$ | $66.7_{\pm 0.0}$ | $52.2_{\pm 7.5}$ | $47.3_{\pm 6.6}$ |

