# OpenReview forum: "AIDBench: A benchmark for evaluating the authorship identification capability of large language models"
_ICLR.cc/2025/Conference — Submitted to ICLR 2025_

### Official Review · Reviewer_8VnX · 2024-10-18

**Soundness:** 2
**Presentation:** 3
**Contribution:** 2
**Rating:** 3
**Confidence:** 4

**Summary:**

This paper introduces a new benchmark, AIDBench, to show the privacy risks posed by large language models (LLMs) in potentially compromising the anonymity of authors across various text formats. The Retrieval-Augmented Generation (RAG)-based method is proposed to help authorship identification for large-scale texts exceeding typical model context windows.

**Strengths:**

1. The authors introduce the RAG approach to address the challenge of one-to-many authorship identification, which could handle large-scale text collections and improve the efficacy of identifying multiple texts by a single author across extensive datasets.
2. The experiments conducted across various text datasets . By demonstrating the model's performance in different scenarios, the authors show the adaptability of the RAG method.

**Weaknesses:**

1. The abstract suggests that the aim of this study is to explore the privacy risks associated with the use of LLMs for recognizing the authorship of anonymous texts. However, the experimental section seems primarily focused on validating the authorship recognition capabilities of these models and proposing new method to enhance their efficiency. There might appear to be a gap as the experiments do not adequately assess the outlined privacy risks, nor do they propose potential mitigation strategies.
2. While the the RAG method is proper for one-to-many authorship identification, providing a comparison with baseline models, particularly under conditions where text lengths do not exceed the model's context window, would enable a better evaluation of the RAG method’s effectiveness.
3. The selection and filtering of datasets, such as the "50 papers filter 10 papers" scenario, might be crucial in evaluating the model's performance but are insufficiently described. A more detailed explanation of how these settings were chosen and optimized are expected.
4. The results section lacks the further analysis of why certain models perform differently under various tests. For instance, the Claude-3.5-sonnet model significantly outperforms others in the task involving 5 authors and 50 papers. It would be insightful if the authors could discuss potential reasons for this model's superior performance and possible factors leading to failures in others.

**Questions:**

The manuscript mentions that embedding models tend to focus more on semantic meanings than on linguistic characteristics. However, there might be instances where texts from the same author share similar linguistic styles but differ significantly in semantic content. How to ensure that the top-k candidates cover texts that belong to the target author in all scenarios?

---

### Official Review · Reviewer_Uj7N · 2024-10-19

**Soundness:** 2
**Presentation:** 3
**Contribution:** 2
**Rating:** 5
**Confidence:** 3

**Summary:**

The paper introduces the AIDBench benchmark that includes a diverse range of datasets to systematically test LLMs’ authorship identification capabilities. The authors test LLMs using two tasks. To enhance LLM performance in authorship identification when texts are too lengthy, the paper introduces a RAG-based method to improve accuracy. This method involves first filtering candidate texts and retrieving the most relevant chunks to keep within the LLM’s context window limits.

**Strengths:**

- The research shows that LLMs, particularly when enhanced by the RAG method, can outperform random chance in identifying authorship, demonstrating potential privacy risks in de-anonymizing texts in various systems.
- The paper introduces a benchmark that includes a variety of datasets from different domains, allowing for a thorough evaluation of LLMs in authorship identification.
- The authors address LLM limitations with long input by introducing a RAG-based method, which improves large-scale authorship identification tasks with large number of candidate authors and text.
- It highlights the underexplored privacy risks posed by LLMs, bringing attention to the potential de-anonymization of authors, particularly in sensitive environments like peer reviews.

**Weaknesses:**

- The paper's methodology for one-to-many identification may have a potential problem. By randomly sampling a number of authors and placing all their writings into a set, there is a risk that all texts in the set come from the same author. This introduces a bias and undermines the validity of the experiment.
- The paper uses identifying anonymous reviewers as one of the motivations, yet the dataset used for analysis consists of research papers rather than actual academic reviews. Consider modify the motivation part, as the two types of text differ substantially in terms of length, tone, and style. Reviews are typically shorter and more opinion-based, while research papers are formal and content-rich.
- In Section 2.4, the paper doesn't clearly discuss whether high cosine similarity is a reliable indicator of authorship. Cosine similarity typically measures the overlap in content or topic rather than writing style. High similarity scores could simply reflect the fact that two texts discuss similar subjects rather than being written by the same author.
- The paper should acknowledge the potential limitation of the benchmark dataset, particularly the likelihood that the data used in this study were also used to train LLMs.
- Section 3.1 notes that the results in Table 2 lead to a different conclusion from prior work. However, this discrepancy may be due to the fact that the previous work employed a different evaluation metric (accuracy) and a different dataset.

**Questions:**

In the one-to-many identification setting, the authors randomly sample a number of authors and compile all of their writings into a set. Does this approach sufficiently simulate real-world conditions? Could this method inadvertently bias the results by creating sets where all texts are from the same author, and if so, how could this be mitigated?

In section 2.4, there is a discussion about using cosine similarity for authorship identification. Should the paper provide more rigorous discussion on whether high cosine similarity actually correlates with authorship? Could high similarity merely indicate shared topics or content rather than distinct writing styles, and if so, how could this be addressed?

Does the paper sufficiently address the potential limitations posed by potential data leakage? What strategies could the authors adopt to mitigate or acknowledge this potential bias in their results?

---

### Official Review · Reviewer_fpKZ · 2024-11-04

**Soundness:** 2
**Presentation:** 3
**Contribution:** 2
**Rating:** 3
**Confidence:** 4

**Summary:**

The authors introduce AIDBench, a new benchmark to evaluate the ability of LLMs to perform authorship identification. The benchmark is made up of five domains and two evaluation settings: one-to-one author identification (authorship verification), and one-to-many author identification (authorship retrieval). Multiple LLMs are prompted, both with and without a topic-controlled prompt, and a RAG-based approach is proposed in cases where the length of the candidate texts exceed context length of the LLMs.

**Strengths:**

•	The research question is interesting, namely whether LLMs can perform authorship identification tasks.

•	The authors collect a new benchmark that could be useful for the community.

**Weaknesses:**

•	It’s not clear whether a new benchmark was needed. There are many authorship verification and authorship retrieval datasets that could’ve been used to evaluate the attribution abilities of LLMs. This would’ve also made comparisons against established methods easier.
* Authorship Verification (One-to-One) –  https://pan.webis.de/clef23/pan23-web/author-identification.html (essays, emails, interviews, and speech transcriptions)
  - https://pan.webis.de/clef22/pan22-web/author-identification.html
  - https://pan.webis.de/clef21/pan21-web/author-identification.html

* Authorship Retrieval (One-to-Many) –
  - Test dataset from - https://arxiv.org/abs/2105.07263

•	It would’ve been better to evaluate with metrics that account for class imbalance, such as the F1 score. This would sharpen the results in Table 2.

•	The RAG-based approach relies on semantic embeddings for it search. Given the nature of the task, it would’ve been more natural to rely upon stylistic embeddings such as the following:
- https://aclanthology.org/2022.repl4nlp-1.26/
- https://aclanthology.org/2021.emnlp-main.70/
- https://arxiv.org/abs/2410.12757

•	Although AIDBench contains 5 domains, it seems that only two domains were evaluated on (in the main text), namely the Guardian and Research Papers.

•	In the One-to-Many scenario, it’s not clear whether one can attribute a research paper to a single author, since they’re collaborative pieces of writing after all. I am not sure what to make of the One-to-Many results in this scenario.

•	There are no baselines compared against. Some of the papers linked above, as well as previous submissions to the PAN CLEF Author Verification challenge would’ve been good baselines.

•	I found the description of the way the one-to-many subsets were created to be confusing. Lines 365-369.

**Questions:**

•	Is it likely that the LLMs have been trained on much of the data of AIDBench? How much does this contribute to their performance?

•	It’s okay not to have the formulas for standard metrics such as precision and recall.

•	Putting the random guess metrics from Table 3 in Table 4 / 5 would’ve made reading those tables much easier.

•	Will the dataset be released?

---

### Meta-Review · Area_Chair_ZnAP · 2024-12-08

**Metareview:**

The paper introduces AIDBench, a benchmark designed to evaluate LLMs in authorship identification tasks. It includes five domains and two evaluation settings: one-to-one (authorship verification) and one-to-many (authorship retrieval). A RAG approach is proposed for handling scenarios where candidate texts exceed LLMs' context limits. Experiments are conducted across multiple datasets and models to assess authorship identification capabilities and explore privacy risks associated with LLMs.

The paper raises an important question about LLMs and authorship identification but fails to justify the need for a new benchmark, provides insufficient baselines and analyses, and misaligns its experiments with its stated privacy goals. Addressing these issues, including leveraging existing datasets, providing clearer experimental setups, and incorporating detailed analyses of privacy risks, would significantly strengthen the work. I recommend rejection in its current form.

**Additional Comments On Reviewer Discussion:**

No author responses.

---

### Decision · Program_Chairs · 2025-01-22

Reject